# Hardness of Noise-Free Learning for Two-Hidden-Layer Neural Networks

**Sitan Chen**
UC Berkeley
sitanc@berkeley.edu

**Aravind Gollakota**
UT Austin
aravindg@cs.utexas.edu

**Adam R. Klivans**
UT Austin
klivans@cs.utexas.edu

**Raghu Meka**
UCLA
raghum@cs.ucla.edu

## Abstract

We give superpolynomial statistical query (SQ) lower bounds for learning two-hidden-layer ReLU networks with respect to Gaussian inputs in the standard (noise-free) model. No general SQ lower bounds were known for learning ReLU networks of any depth in this setting: previous SQ lower bounds held only for adversarial noise models (agnostic learning) [KK14, GGK20, DKZ20] or restricted models such as correlational SQ [GGJ+20, DKKZ20]. Prior work hinted at the impossibility of our result: Vempala and Wilmes [VW19] showed that general SQ lower bounds cannot apply to any real-valued family of functions that satisfies a simple non-degeneracy condition. To circumvent their result, we refine a lifting procedure due to Daniely and Vardi [DV21] that reduces Boolean PAC learning problems to Gaussian ones. We show how to extend their technique to other learning models and, in many well-studied cases, obtain a more efficient reduction. As such, we also prove new cryptographic hardness results for PAC learning two-hidden-layer ReLU networks, as well as new lower bounds for learning constant-depth ReLU networks from label queries.

## 1  Introduction

In this paper we extend a central line of research proving representation-independent hardness results for learning classes of neural networks. We will consider arguably the simplest possible setting: given samples $(x_1, y_1), \ldots, (x_n, y_n)$ where for every $i \in [n]$, $x_i$ is sampled independently from some distribution $\mathcal{D}$ over $\mathbb{R}^d$ and $y_i = f(x_i)$ for an unknown neural network $f : \mathbb{R}^d \to \mathbb{R}$, the goal is to output any function $\widehat{f}$ for which $\mathbb{E}_{x \sim \mathcal{D}}[(f(x) - \widehat{f}(x))^2]$ is small. This model is often referred to as the *realizable* or *noise-free* setting.

This problem has long been known to be computationally hard for discrete input distributions. For example, if $\mathcal{D}$ is supported over a discrete domain like the Boolean hypercube, then we have a variety of hardness results based on cryptographic/average-case assumptions [KS09, DLSS14, DSS16, DV20, DV21].

Over the last few years there has been a very active line of research on the complexity of learning with respect to continuous distributions, the most widely studied case being the assumption that $\mathcal{D}$ is a standard Gaussian in $d$ dimensions. A rich algorithmic toolbox has been developed for the Gaussian setting [JSA15, ZSJ+17, BG17, LY17, Tia17, GKM18, GLM18, BJW19, ZYWG19, DGK+20, LMZ20, DK20, ATV21, CKM20, SZB21, VSS+22], but all known efficient algorithms can only handle networks with a single hidden layer, that is, functions of the form $f(x) = \sum_{i=1}^{k} \lambda_i \sigma(\langle w_i, x \rangle)$.

36th Conference on Neural Information Processing Systems (NeurIPS 2022).

This motivates the following well-studied question:

*Are there fundamental barriers to learning neural networks with* two *hidden layers?* (1)

Two distinct lines of research, one using cryptography and one using the statistical query (SQ) model, have made progress towards solving this question.

In the cryptographic setting, [DV21] showed that the existence of a certain class of pseudorandom generators, specifically local pseudorandom generators with polynomial stretch, implies superpolynomial lower bounds for learning ReLU networks with *three* hidden layers.

For SQ learning, work of [GGJ+20] and [DKKZ20] gave the first superpolynomial *correlational* SQ (CSQ) lower bounds for learning even one-hidden-layer neural networks. Notably, however, there are strong separations between SQ and CSQ [APVZ14, ADHV19, CKM20], and the question of whether a general SQ algorithm exists remained an interesting open problem. In fact, Vempala and Wilmes [VW19] showed that general SQ lower bounds might be impossible to achieve for learning real-valued neural networks. For any family of networks satisfying a simple non-degeneracy condition (see Section 1.1), they gave an algorithm that succeeded using only polynomially many statistical queries. As such, the prevailing conventional wisdom was that noise was required in the model to obtain full SQ lower bounds.

The main contribution of this paper is to answer Question 1 by giving both general SQ lower bounds and cryptographic hardness results (based on the Learning with Rounding or LWR assumption) for learning ReLU networks with two hidden layers and polynomially bounded weights. We note that our SQ lower bound is the first of its kind for learning ReLU networks of *any* depth. We also show how to extend our results to the setting where the learner has label query access to the unknown network.

| Reference | Num. hidden layers | Model of hardness |
|---|---|---|
| [DKKZ20, GGJ+20] | 1 | Correlational SQ |
| [DV21] | 3 | Cryptographic (assuming existence of local PRGs) |
| *This work* | 2 | Full SQ |
| *This work* | 2 | Cryptographic (assuming hardness of LWR) |

Table 1: Summary of known and new superpolynomial lower bounds for learning noise-free shallow ReLU networks over Gaussian inputs up to sufficiently small (but non-negligible) error. (Definitions and terminology may be found in Appendix A.)

**SQ Lower Bound**    We state an informal version of our main SQ lower bound:

**Theorem 1.1** (Full SQ lower bound for two hidden layers (informal), see Theorem 3.1)**.** *Any SQ algorithm for learning* $\mathrm{poly}(d)$*-sized two-hidden-layer ReLU networks over* $\mathcal{N}(0, \mathrm{Id}_d)$ *up to squared loss* $1/\mathrm{poly}(d)$ *must use at least* $d^{\omega(1)}$ *queries, or have query tolerance that is negligible in* $d$.

We stress that this bound holds unconditionally, independent of any cryptographic assumptions. This simultaneously closes the gap between the hardness result of [DV21] and the positive results on one-hidden-layer networks [JSA15, ZSJ+17, GLM18, ATV21, DK20] and goes against the conventional wisdom that one cannot hope to prove full SQ lower bounds for learning real-valued functions in the realizable setting.

We also note that unlike previous CSQ lower bounds which are based on orthogonal function families and crucially exploit cancellations specific to the Gaussian distribution, our Theorem 1.1 and other hardness results in this paper extend to any reasonably anticoncentrated product distribution over $\mathbb{R}^d$; see Remark C.5.

**Cryptographic Lower Bound**    While Theorem 1.1 rules out most known approaches for provably learning neural networks (e.g. method of moments/tensor decomposition [JSA15, ZSJ+17,

GLM18, BJW19, DGK$^+$20, DK20, ATV21], noisy gradient descent [BG17, LY17, Tia17, GKM18, ZYWG19, LMZ20], and filtered PCA [CKM20]), it does not preclude the existence of a non-SQ algorithm for doing so. Indeed, a number of recent works [BRST21, SZB21, ZSWB22, DK21] have ported algorithmic techniques like lattice basis reduction [LLL82], traditionally studied in the context discrete settings like cryptanalysis, to learning problems over continuous domains for which there is no corresponding SQ algorithm.

Our next result shows however that under a certain cryptographic assumption, namely hardness of *Learning with Rounding (LWR) with polynomial modulus* [BPR12, AKPW13, BGM$^+$16], no polynomial-time algorithm can learn two-hidden-layer neural networks from Gaussian examples. The LWR problem is a close cousin of the well-known Learning with Errors (LWE) problem [Reg09], except with deterministic rounding in place of random additive errors.

**Definition 1.2.** Fix moduli $p, q \in \mathbb{N}$, where $p < q$, and let $n$ be the security parameter. For any $w \in \mathbb{Z}_q^n$, define $f_w : \mathbb{Z}_q^n \to \mathbb{Z}_p/p$ by $f_w(x) = \frac{1}{p}\lfloor w \cdot x \rceil_p = \frac{1}{p}\lfloor \frac{p}{q}(w \cdot x \bmod q)\rceil$, where $\lfloor t \rceil$ is the closest integer to $t$. In the $\mathsf{LWR}_{n,p,q}$ problem, the secret $w$ is drawn randomly from $\mathbb{Z}_q^n$, and we must distinguish between labeled examples $(x, y)$ where $x \sim \mathbb{Z}_q^n$ and either $y = f_w(x)$ or $y$ is drawn independently from $\mathrm{Unif}(\mathbb{Z}_p/p)$. LWE is similar, except that $y \in \mathbb{Z}_q/q$ is either $\frac{1}{q}((w \cdot x + e) \bmod q)$ for some $e \in \mathbb{Z}_q$ sampled from a carefully chosen noise distribution, or is drawn from $\mathrm{Unif}(\mathbb{Z}_q/q)$.

**Theorem 1.3** (Cryptographic hardness result (informal), see Theorem 4.1). *Suppose there exists a* $\mathrm{poly}(d)$*-time algorithm for learning* $\mathrm{poly}(d)$*-sized two-hidden-layer ReLU networks over* $\mathcal{N}(0, \mathrm{Id}_d)$ *up to squared loss* $1/\mathrm{poly}(d)$*. Then there exists a quasipolynomial-time algorithm for* LWR *with polynomial modulus (i.e., in the regime where* $n = d$*,* $p, q = \mathrm{poly}(n)$*, and* $q/p = \mathrm{poly}(n)$*).*

Note that here we may actually improve the LWR hardness assumption required from quasipolynomial to any mildly superpolynomial function of the security parameter (see Remark 4.2).

Under LWR with polynomial modulus, we also show the first hardness result for learning *one hidden layer* ReLU networks over the uniform distribution on $\{0, 1\}^d$ (see Theorem 4.3).

We discuss existing hardness evidence for LWR as well as its relation to more standard assumptions like LWE in Appendix A.3. From a negative perspective, Theorem 1.3 suggests that the aforementioned lattice-based algorithms for continuous domains are unlikely to yield new learning algorithms for two-hidden-layer networks, because even their more widely studied *discrete* counterparts have yet to break LWR. From a positive perspective, in light of the prominent role LWR and its variants have played in a number of practical proposals for post-quantum cryptography [CKLS18, BGML$^+$18, JZ16, DKRV18], Theorem 1.3 offers a new avenue for stress-testing these schemes.

**Query Learning Lower Bound**  One additional benefit of our techniques is that they are flexible enough to accommodate other learning models beyond traditional PAC learning. To illustrate this, for our final result we show hardness of learning neural networks from *label queries*. In this setting, the learner is much more powerful: rather than sample or SQ access, they are given the ability to query the value $f(x)$ of the unknown function $f$ at any desired point $x$ in $\mathbb{R}^d$, and the goal is still to output a function $\widehat{f}$ for which $\mathbb{E}[(f(x) - \widehat{f}(x))^2]$ is small. The expectation here is with respect to some specified distribution, which we will take to be $\mathcal{N}(0, \mathrm{Id}_d)$.

In recent years, this question has received renewed interest from the security and privacy communities in light of *model extraction attacks*, which attempt to reverse-engineer neural networks found in publicly deployed systems [TJ$^+$16, MSDH19, PMG$^+$17, JCB$^+$20, RK20, JWZ20, DG21]. Recent work [CKM21] has shown that in this model, there is an efficient algorithm for learning arbitrary one-hidden-layer ReLU networks that is truly polynomial in all relevant parameters. We show that under plausible cryptographic assumptions about the existence of simple pseudorandom function (PRF) families (see Section 5) which may themselves be based on standard number theoretic or lattice-based cryptographic assumptions, such a guarantee is impossible for general *constant-depth* ReLU networks.

**Theorem 1.4** (Label query hardness (informal), see Theorem 5.1). *If either the decisional Diffie–Hellman or the Learning with Errors assumption holds, then the class of* $\mathrm{poly}(d)$*-sized constant-depth ReLU networks from* $\mathbb{R}^d$ *to* $\mathbb{R}$ *is not learnable up to small constant squared loss* $\varepsilon$ *over* $\mathcal{N}(0, \mathrm{Id}_d)$ *even using label queries over all of* $\mathbb{R}^d$*.*

Note that the connection between PRFs and hardness of learning from label queries over *discrete domains* is a well-known connection dating back to Valiant [Val84]. To our knowledge, however, Theorem 1.4 is the first hardness result for query learning over continuous domains.

## 1.1 Discussion and Related Work

**Hardness for learning neural networks.** There are a number of works [BR89, Vu06, KS09, LSSS14, GKKT17, DV20] showing hardness for *distribution-free* learning of various classes of neural networks.

As for hardness of distribution-specific learning, several works have established lower bounds with respect to the Gaussian distribution. Apart from the works [GGJ+20, DKKZ20, DV21] from the introduction which are most closely related to the present work, we also mention the works of [KK14, GKK19, GGK20, DKZ20] which showed hardness for *agnostically* learning halfspaces and ReLUs, [Sha18] which showed hardness for learning periodic activations with gradient-based methods, [SVWX17] which showed lower bounds against SQ algorithms for learning one-hidden-layer networks using *Lipschitz* statistical queries and large tolerance, and [SZB21] which showed lattice-based hardness of learning one-hidden-layer networks when the labels $y_i$ have been perturbed by bounded *adversarially chosen* noise. Our approach has similarities to the "Gaussian lift" as studied by Klivans and Kothari [KK14]. Their approach, however, required noise in the labels, whereas we are interested in hardness in the strictly *realizable setting*. We also remark that [DGKP20, AAK21] showed *correlational* SQ lower bounds for learning random depth-$\omega(\log n)$ neural networks over *Boolean inputs* which are uniform over a halfspace.

There have also been works on hardness of learning from label queries over *discrete domains* and for more "classical" concept classes like Boolean circuits [Fel09, CGV15, Val84, Kha95, AK95].

**SQ lower bounds for real-valued functions.** A recurring conundrum in the literature on SQ lower bounds for supervised learning has been whether one can show SQ hardness for learning *real-valued* functions. SQ lower bounds for Boolean functions are typically shown by lower bounding the *statistical dimension* of the function class, which essentially corresponds to the largest possible set of functions in the class which are all approximately pairwise orthogonal. Indeed, the content of the hardness results of [GGJ+20, DKKZ20] was to prove lower bounds on the statistical dimension of one-hidden-layer networks. Unfortunately, for real-valued functions, statistical dimension lower bounds only imply CSQ lower bounds. As discussed in [GGJ+20], the class of $d$-variate Hermite polynomials of degree-$\ell$ is pairwise orthogonal and of size $d^{O(\ell)}$, which translates to a CSQ lower bound of $d^{\Omega(\ell)}$. Yet there exist SQ algorithms for learning Hermite polynomials in far fewer queries [APVZ14, ADHV19].

Further justification for the difficulty of proving SQ lower bounds for real-valued functions came from [VW19], which observed that for any real-valued learning problem satisfying a seemingly innocuous non-degeneracy assumption—namely that for any pair of functions $f, g$ in the class, the probability under the input distribution $\mathcal{D}$ that $f(x) = g(x)$ is zero—there is an efficient "cheating" SQ algorithm (see Proposition 4.1 therein). The SQ lower bound shown in the present work circumvents this proof barrier by exhibiting a family of neural networks for which any pair of networks agrees on a set of inputs with Gaussian measure *bounded away from zero*.

**Open question.** All known positive results for one hidden layer that run in time polynomial in all parameters require various assumptions on the underlying network. This leaves open the tantalizing possibility of strengthening our results to apply to worst-case one hidden layer networks.

## 1.2 Technical Overview

Our work will build on a recent approach of Daniely and Vardi [DV21], who developed a simple and clever technique for lifting discrete functions to the Gaussian domain entirely in the realizable setting. Our main contributions are to (1) make their lifting procedure more efficient so that two hidden layers suffice and (2) show how to apply the lift in a variety of models beyond PAC. For the purposes of this overview we will take the domain of our discrete functions to be $\{0, 1\}^d$, but our techniques extend to $\mathbb{Z}_q^d$ with $q = \text{poly}(d)$.

**Daniely–Vardi (DV) lift.** At a high level, the DV lift is a transformation mapping a Boolean example $(x, y)$ labeled by a hard-to-learn Boolean function $f$ to a Gaussian example $(z, \widetilde{y})$ labeled by a (real-valued) ReLU network $f^{\mathsf{DV}}$ that behaves similarly to $f$ in that $f^{\mathsf{DV}}(z)$ approximates $f(\text{sign}(z))$, where for us $\text{sign}(t)$ denotes $\mathbb{1}[t > 0]$ and is applied elementwise. The key idea is to use a continuous approximation $\widetilde{\text{sign}}$ of the $\text{sign}$ function, and to pair it with a "soft indicator" function $\text{bad} : \mathbb{R}^d \to \mathbb{R}_+$ that is large whenever $\text{sign}(z) \neq \widetilde{\text{sign}}(z)$, and that can be implemented as a one-hidden-layer network independent of the target function. One can show that whenever $f$ is realizable as an $L$-hidden-layer network over $\{0, 1\}^d$, the function $f^{\mathsf{DV}}(z) = \text{ReLU}(f(\widetilde{\text{sign}}(z)) - \text{bad}(z))$ can be implemented as an $(L + 2)$-hidden-layer network satisfying

$$f^{\mathsf{DV}}(z) = \text{ReLU}(f(\text{sign}(z)) - \text{bad}(z)).$$

This property allows us to generate synthetic Gaussian labeled examples $(z, f^{\mathsf{DV}}(z))$ from Boolean labeled examples $(x, f(x))$, and thereby reduce the problem of learning $f$ to that of learning $f^{\mathsf{DV}}$.

**Improving the DV lift.** Our first technical contribution is to introduce a more efficient lift which only requires one extra hidden layer. Our starting point is to observe that a variety of hard-to-learn Boolean functions $f$ like parity and $\mathsf{LWR}$ take the form $f(x) = \sigma(h(x))$ for some ReLU network $h$ whose range $T$ over Boolean inputs is a discrete subset of $[0, \text{poly}(d)]$ of *polynomially bounded* size, and for some function $\sigma : T \to [0, 1]$. For such *compressible* functions (see Definition 2.1), one can write $f(x) = \sigma(h(x)) = \sum_{t^* \in T} \sigma(t^*) \mathbb{1}[h(x) = t^*]$. Again, we would like to implement lifted function $f^{\triangle} : \mathbb{R}^d \to \mathbb{R}$ using $\widetilde{\text{sign}}$ and $\text{bad}$ so that it approximates $f(\text{sign}(z))$ except when $\text{bad}$ indicates that $\widetilde{\text{sign}} \neq \text{sign}$. To this end, we might hope to implement, say,

$$f^{\triangle}(z) = \sum_{t^* \in T} \sigma(t^*) \mathbb{1}[h(\widetilde{\text{sign}}(z)) = t^*] \mathbb{1}[\forall j : \text{bad}(z_j) \ll 1].$$

Here we now view $\text{bad}$ as a univariate function, and whenever it is small, we can be sure $\widetilde{\text{sign}} = \text{sign}$. Suppose that we could build a one-hidden-layer network $N(s_1, \ldots, s_d; t)$ that behaves like $\mathbb{1}[t = 0]\mathbb{1}[\forall j : s_j \ll 1]$. Then we could realize $f^{\triangle}$ as an $(L + 1)$-hidden-layer network:

$$f^{\triangle}(z) = \sum_{t^* \in T} \sigma(t^*) N(\text{bad}(z_1), \ldots, \text{bad}(z_d); h(\widetilde{\text{sign}}(z)) - t^*).$$

While many natural attempts to build such an $N$ run into difficulties, we construct a suitably relaxed version of $N$ that turns out to suffice for the reduction. To gain some intuition for our construction, the starting observation is that the following inclusion-exclusion type formula vanishes identically whenever any of the $s_j$ is 1:

$$\psi(s_1, s_2, s_3) - \psi(1, s_2, s_3) - \psi(s_1, 1, s_3) - \psi(s_1, s_2, 1)$$
$$+ \psi(s_1, 1, 1) + \psi(1, s_2, 1) + \psi(s_1, 1, 1) - \psi(1, 1, 1).$$

For a suitable choice of $\psi$, one might hope to build $N$ out of such a formula by taking $s_j = \text{bad}(z_j)$ for every $j$. But the natural generalization of this expression to $d$ inputs would have size $2^d$, which runs the risk of rendering the resulting SQ lower bounds vacuous. Our final construction (Lemma 2.6) instead resembles a truncated inclusion-exclusion type formula of only quasipolynomial size, which may be of independent interest. Since the SQ lower bounds for Boolean functions that we build on are exponential, by a simple padding argument we still obtain a superpolynomial SQ lower bound for our lifted functions.

**Hard one-hidden-layer Boolean functions and $\mathsf{LWR}$.** To use this lift for Theorems 1.1 and 1.3, we need one-hidden-layer networks that are *compressible* and hard to learn over uniform Boolean inputs. For SQ lower bounds, we can simply start from parities, for which there are exponential SQ lower bounds, and which turn out to be easily implementable by compressible one-hidden-layer networks. For cryptographic hardness, Daniely and Vardi [DV21] used certain one-hidden-layer Boolean networks that arise from the cryptographic assumption that local PRGs exist (see Section A.4.1 therein). Unfortunately, these functions are not compressible. For this reason, we work instead with $\mathsf{LWR}$: it turns out that the $\mathsf{LWR}$ functions are compressible and, conveniently, the hardness assumption directly involves uniform discrete inputs.

**Hardness beyond PAC.** While the DV lift is *a priori* only for showing hardness of example-based PAC learning, we can extend it to the SQ and label query models by simple simulation arguments.

## 2  Compressing the Daniely–Vardi Lift

In this section we show how to refine the lifting procedure of Daniely and Vardy [DV21] such that whenever the underlying discrete functions satisfy a property we term *compressibility*, we obtain hardness under the Gaussian for networks with just one extra hidden layer.

**Definition 2.1.** Let $q > 0$ be a modulus.[1] We call an $L$-hidden-layer ReLU network $f : \mathbb{Z}_q^d \to [0, 1]$ *compressible* if it is expressible in the form $f(x) = \sigma(h(x))$, where

- $h : \mathbb{Z}_q^d \to T$ is an $(L-1)$-hidden-layer network such that $|h(x)| \leq \text{poly}(d)$ for all $x$;
- $h$ has range $T = h(\mathbb{Z}_q^d)$ such that $T \subseteq \mathbb{Z}$ and $|T| \leq \text{poly}(d)$; and
- $\sigma : T \to [0, 1]$ is a mapping from $h$'s possible output values to $[0, 1]$.

*Remark* 2.2. To see why such an $f$ is an $L$-hidden-layer network in $z$, consider the function $\sigma : T \to \mathbb{R}$. Because $T \subseteq \mathbb{Z}$ and $|T| \leq \text{poly}(d)$, $\sigma$ is expressible as (the restriction to $T$ of) a piecewise linear function on $\mathbb{R}$ whose size and maximum slope are $\text{poly}(d)$, and hence as a $\text{poly}(d)$-sized one-hidden-layer ReLU network from $\mathbb{R}$ to $\mathbb{R}$. By composition, $x \mapsto \sigma(h(x))$ can be represented by an $L$-hidden-layer network.

We now formally state a theorem which captures our "compressed" version of the DV lift. The version of this theorem for $L + 2$ layers is implicit in [DV21]. In technical terms, our improvement consists of removing the single outer ReLU present in their construction. Thus, while our construction still has three *linear* layers, it has only two *non-linear* layers. By a standard padding argument, we also obtain Corollary C.6, which lets us work with polynomial-sized neural networks.

**Theorem 2.3** (Compressed DV lift). *Let $q = \text{poly}(d)$ be a modulus. Let $\mathcal{C}$ be a class of compressible $L$-hidden-layer $\text{poly}(d)$-sized ReLU networks mapping $\mathbb{Z}_q^d$ to $[0, 1]$. Let $m = m(d) = \omega_d(1)$ be a size parameter that grows slowly with $d$. There exists a class $\mathcal{C}^\triangle$ of $(L+1)$-hidden-layer $d^{\Theta(m)}$-sized ReLU networks mapping $\mathbb{R}^d$ to $[0, 1]$ such that the following holds:*

*Suppose there is an efficient algorithm $A$ capable of learning $\mathcal{C}^\triangle$ over $\mathcal{N}(0, \text{Id}_d)$ up to squared loss $d^{-\Theta(m)}$. Then there is an efficient algorithm $B$ capable of weakly predicting $\mathcal{C}$ over $\text{Unif}(\mathbb{Z}_q^d)$ with advantage $d^{-\Theta(m)}$ over guessing the constant $1/2$ in the following sense: given access to labeled examples $(x, f(x))$ for $x \sim \text{Unif}(\mathbb{Z}_q^d)$ and an unknown $f \in \mathcal{C}$, $B$ satisfies $\mathbb{E}\left[(B(x) - f(x))^2\right] < \mathbb{E}\left[\left(\frac{1}{2} - f(x)\right)^2\right] - d^{-\Theta(m)}$, where the probability is taken over both $x$ and the internal randomness of $B$. We refer to $\mathcal{C}^\triangle$ as the* lifted class *corresponding to $\mathcal{C}$.*

The proof of Theorem 2.3 leverages certain one-hidden-layer gadgets. The first two gadgets are inherent to the original DV lift (extended to work with general $\mathbb{Z}_q$ as opposed to just $\{0, 1\}$), while the third is one of our main technical contributions and essential to obtaining an improvement in depth. Proofs are deferred to Appendix C.

Start by letting $I_0, I_1, \ldots, I_{q-1}$ be a partition of $\mathbb{R}$ into $q$ consecutive intervals each of mass $1/q$ under $\mathcal{N}(0, 1)$ (e.g., when $q = 2$, $I_0 = (-\infty, 0)$ and $I_1 = (0, \infty)$). Note that these intervals will have differing lengths, which we denote by $|I_j|$, and the shortest ones will be the ones closest to the origin. Still, by Gaussian anti-concentration, we know that each $|I_j| \geq \Theta(1/q)$. Let $\text{thres}_q : \mathbb{R} \to \mathbb{Z}_q$ be the piecewise constant function that takes on value $k$ on $I_k$. Clearly, when $t \sim \mathcal{N}(0, 1)$, $\text{thres}_q(t) \sim \text{Unif}(\mathbb{Z}_q)$. Let $R_1, \ldots, R_q$ be intervals such that $R_k \subseteq I_{k-1} \cup I_k$ and $R_k$ contains the boundary point between $I_{k-1}$ and $I_k$, and such that each $R_k$ has mass $\delta/q$ for some $\delta \ll 1$ to be picked later. Let $S_1, \ldots, S_q$ be slightly larger intervals such that $R_k \subset S_k$ for each $k \in [q-1]$, and each $S_k$ has mass $2\delta/q$. By Gaussian anti-concentration again, each $|S_k| \geq \Theta(\delta/q)$. Notice that by construction, $\mathbb{P}_{z \sim \mathcal{N}(0,1)}[z \in \cup_k R_k] = \delta$ and $\mathbb{P}_{z \sim \mathcal{N}(0,1)}[z \in \cup_k S_k] = 2\delta$.

**Lemma 2.4.** *Let $\delta > 0$, $q > 0$, and intervals $I_k, R_k, S_k$ for $k \in \mathbb{Z}_q$ be as above. There exists a one-hidden-layer ReLU network $N_1 : \mathbb{R} \to \mathbb{R}$ with $O(q)$ units and weights of magnitude $O(q/\delta)$ such that $N_1(t) = \text{thres}_q(t)$ if $t \notin \cup_k R_k$.*

---

[1]Our results are stronger when $q$ is taken to be a large polynomial in the dimension, but the Boolean $q = 2$ case is illustrative of all the main ideas.

**Lemma 2.5.** *Let $\delta > 0$, $q > 0$, and intervals $I_k, R_k, S_k$ for $k \in \mathbb{Z}_q$ be as above. There exists a one-hidden-layer ReLU network $N_2 : \mathbb{R} \to [0, 1]$ with $O(q)$ units and weights of magnitude $O(q/\delta)$ such that*

$$
N_2(t) \text{ is } \begin{cases} = 1 & \text{if } t \in \cup_k R_k \\ = 0 & \text{if } t \in \mathbb{R} \setminus \cup_k S_k \\ \geq 0 & \text{otherwise} \end{cases} .
$$

Note that when $q = 2$, $N_1$ and $N_2$ play the role of "$\widetilde{\text{sign}}$" and "$\text{bad}$" from the technical overview.

To motivate the third gadget, recall from the technical overview that one might hope to build $N_3(s_1, \ldots, s_d; t)$ that behaves like $\mathbb{1}[t = 0]\mathbb{1}[\forall j : s_j \ll 1]$. Slightly more generally, one can show that it would suffice to build a one-hidden-layer network $N_3$ with the following properties:

$$
N_3(s_1, \ldots, s_d; t) = \begin{cases} 0 & \text{if } \exists j : s_j = 1 \\ 0 & \text{if } t \in \mathbb{Z} \setminus \{0\} \\ 1 & \text{if } \forall j : s_j = 0 \text{ and } t = 0 \end{cases} \tag{2}
$$

Unfortunately, most natural attempts to construct $N_3$ with such ideal properties run into difficulties and appear to require *two* hidden layers (see Appendix D for discussion).

The key idea that lets us make progress is to restrict attention to those possibilities for $(s_1, \ldots, s_d) = (N_2(z_1), \ldots, N_2(z_d))$ that are the most likely. Specifically, if $m = \omega_d(1)$ is the size parameter from Theorem 2.3, then by setting $\delta$ in Lemmas 2.4 and 2.5 appropriately, we can ensure that with overwhelming probability over $z \sim \mathcal{N}(0, \text{Id})$, no more than $m$ of the $N_2(z_j)$ are simultaneously 1. Accordingly, we focus on constructing $N_3$ such that

$$
N_3(s_1, \ldots, s_d; t) = \begin{cases} 0 & \text{if between 1 and } m \text{ of the } s_i \text{ are 1} \\ 0 & \text{if } t \in \mathbb{Z} \setminus \{0\} \\ 1 & \text{otherwise} \end{cases} . \tag{3}
$$

Our construction for $N_3$ has size $d^{\Theta(m)}$, and satisfies the first and second properties exactly. It also "approximately" satisfies the third in the sense that it takes on a nonzero value with nonnegligible probability over its inputs. As we will see, this turns out to be enough for the reduction to go through. And even though the size of $N_3$ is slightly superpolynomial in the dimension, because the SQ lower bounds for Boolean functions that we build on are exponential, by a simple padding argument we will still obtain a superpolynomial SQ lower bound for our lifted functions.

**Lemma 2.6** (Main lemma). *Let $m = m(d) = \omega_d(1)$ be a size parameter. There exists a one-hidden-layer neural network $N_3 : \mathbb{R}^d \times \mathbb{R} \to \mathbb{R}$ such that*

- (a) $N_3(s_1, \ldots, s_d; t) = 0$ *for any $t \in \mathbb{R}$ if between 1 and $m$ of the $s_j$ are 0*
- (b) $N_3(s_1, \ldots, s_d; t) = 0$ *for any $s_1, \ldots, s_d \in [0, 1]^d$ if $t \in \mathbb{Z} \setminus \{0\}$*
- (c) $N_3$ *has size at most $d^{2m}$*
- (d) $N_3(0, \ldots, 0, s; 0) = s$ *for any $s \in [0, \frac{1}{d}]$ (there are $d - 1$ zeroes in front of s).*

*Proof sketch of Theorem 2.3.* For each $f \in \mathcal{C}$ given by $f = \sigma \circ h$, let $f^\triangle \in \mathcal{C}^\triangle$ be given by

$$
f^\triangle(z) = \sum_{t^* \in T} \sigma(t^*) N_3(N_2(z_1), \ldots, N_2(z_d); h(N_1(z)) - t^*), \tag{4}
$$

where $N_1$ and $N_2$ are from Lemmas 2.4 and 2.5, with the $\delta$ parameter set to $d^{-10m}$, and $N_3$ is from Lemma 2.6. This is an $(L+1)$-hidden layer network since $h \circ N_1$ and $N_2$ each have at most $L$ hidden layers, and $N_3$ adds an additional layer. By Lemma 2.6(c), the size of this network is $S = d^{\Theta(m)}$. Note that for $z$ such that $N_2(z_1), \ldots, N_2(z_d) < 1$, we have $N_1(z) = \text{thres}_q(z)$, and the only $t^*$ for which one of the summands in Eq. (4) is potentially nonzero is the one given by $t^* = h(\text{thres}_q(z))$. So in this case $f^\triangle$ simplifies to

$$
f^\triangle(z) = f(\text{thres}_q(z)) \, N_3(N_2(z_1), \ldots, N_2(z_d); 0). \tag{5}
$$

Further, for $z$ such that between 1 and $m$ of the $N_2(z_j)$ are 1, we know that $\psi(N_2(z_1), \ldots, N_2(z_d); t) = 0$ identically (for all $t \in \mathbb{R}$), so in this case $f^\triangle(z) = 0$. And finally, for

$z$ such that more than $m$ of the $N_2(z_j)$ are 1, we have no guarantees on the behavior of $f^\triangle$, but as we now show, we have set parameters such that this case occurs only with negligible probability, and we can pretend that 0 is still a valid label in this case. Indeed, by standard Gaussian anti-concentration, for each coordinate $z_j$ we have $\mathbb{P}_{z_j}[N_2(z_j) = 1] = \mathbb{P}_{z_j}[z_j \in \cup_k R_k] = \delta = d^{-10m}$. The number of coordinates $j$ for which $N_2(z_j) = 1$ thus follows a binomial distribution $B(d, d^{-10m})$, which has a decreasing pdf with unique mode at $\lfloor (d+1)d^{-10m} \rfloor = 0$. Thus the probability of having at least $m$ 1s is at most

$$\sum_{i=m}^d \binom{d}{i}(d^{-10m})^i(1 - d^{-10m})^{d-i} \le (d - m + 1)\binom{d}{m}d^{-10m^2} \le dd^m d^{-10m^2} \le d^{-9m^2} \quad (6)$$

for sufficiently large $d$. This is negligibly small not only in $d$ but also in $S = d^{\Theta(m)}$.

We now describe the reduction. For each labeled example $(x, y)$ that the discrete learner $B$ receives, where $x \sim \mathrm{Unif}(\mathbb{Z}_q^d)$ and $y = f(x)$ for an unknown $f \in \mathcal{C}$, $B$ forms a labeled example $(z, \widetilde{y})$ for the Gaussian learner $A$ as follows. For each coordinate $j \in [d]$, $z_j$ is drawn from $\mathcal{N}(0, 1)$ conditioned on $z_j \in I_{x_j}$. Notice that this way $\mathrm{thres}_q(z) = x$, and the marginal distribution on $z$ is exactly $\mathcal{N}_d$. The modified label is given by

$$\widetilde{y} = \widetilde{y}(y, z) = \begin{cases} 0 & \text{if more than } m \text{ of the } N_2(z_j) \text{ are 1} \\ 0 & \text{if between 1 and } m \text{ of the } N_2(z_j) \text{ are 1} \\ y\, N_3(N_2(z_1), \ldots, N_2(z_d);\, 0) & \text{otherwise} \end{cases} \quad (7)$$

Note that in the bottom two cases, $\widetilde{y} = f^\triangle(z)$ exactly; in the top case $\widetilde{y}$ is in general inconsistent with $f^\triangle$, but as we have seen, this case occurs with $\mathrm{negl}(S)$ probability. In particular, with overwhelming probability, no $\mathrm{poly}(S)$-time algorithm will ever see non-realizable samples.

So $B$ can feed these new labeled examples $(z, \widetilde{y})$ to $A$. Suppose $A$ outputs a hypothesis $\widehat{f} : \mathbb{R}^d \to \mathbb{R}$ such that $\mathbb{E}_{z \sim \mathcal{N}_d}[(\widehat{f}(z) - f^\triangle(z))^2] \le \varepsilon$. We need to show $B$ can convert this hypothesis into a nontrivial one for its discrete problem. We first define a "good region" $G \subseteq \mathbb{R}^d$ where $f^\triangle$ is guaranteed to be nonzero and nontrivially related to the original $f$ by saying $z \in G$ iff $N_2(z_1), \ldots, N_2(z_{d-1}) = 0$, and $N_2(z_d) \in (\frac{1}{2d}, \frac{1}{d})$. Observe that when $z \in G$, by Eq. (5) and Lemma 2.6(d) we have

$$\begin{aligned} f^\triangle(z) &= f(\mathrm{thres}_q(z))N_3(N_2(z_1), \ldots, N_2(z_{d-1}), N_2(z_d); 0) \\ &= f(x)N_3(0, \ldots, 0, N_2(z_d); 0) \\ &= yN_2(z_d), \end{aligned} \quad (8)$$

where we use the fact that $\mathrm{thres}_q(z) = x$, so that $f(\mathrm{thres}_q(z)) = f(x) = y$.

One can show that $G$ has non-negligible probability mass. The discrete learner $B$ can now adapt $\widehat{f}$ as follows. Given a fresh test point $x \sim \mathrm{Unif}(\mathbb{Z}_q^d)$, the learner forms $z = z(x)$ such that for each coordinate $j \in [d]$, $z_j$ is drawn from $\mathcal{N}(0, 1)$ conditioned on $z_j \in I_{x_k}$. If $z \in G$, then $B$ predicts $\widehat{y} = \frac{\widehat{f}(z)}{N_2(z_d)}$ (recall that when $z \in z$, $N_2(z_d) > \frac{1}{2d}$), and otherwise it simply predicts $\widetilde{y} = \frac{1}{2}$. By exploiting the fact that this is a good prediction at least on the region $G$, it is not hard to show that $B$'s overall square loss is non-negligibly better than random. $\square$

## 3 Statistical Query Lower Bound

We prove a superpolynomial SQ lower bound (for general queries as opposed to only correlational or Lipschitz queries) for weakly learning two-hidden-layer ReLU networks under the standard Gaussian. We obtain this by lifting the problem of learning parities under $U_d$, which is well-known to require exponentially many queries.

**Theorem 3.1.** *Fix any $\alpha \in (0, 1)$. Any SQ learner capable of learning $\mathrm{poly}(d)$-sized two-hidden-layer ReLU networks under $\mathcal{N}(0, \mathrm{Id}_d)$ up to squared loss $\varepsilon$ (for some sufficiently small $\varepsilon = 1/\mathrm{poly}(d)$) using bounded queries of tolerance $\tau \ge 2^{-(\log d)^{2-\alpha}}$ must use at least $\Omega(2^{2^{(\log d)^\alpha}} \tau^2) = d^{\omega(1)}\tau^2$ such queries.*

This theorem is proven using the following key reduction, which adapts the compressed DV lift (Theorem 2.3) to the SQ setting. The proof is deferred to Appendix E.

**Theorem 3.2.** *Let $q = \mathrm{poly}(d)$ be a modulus, and let $m = m(d) = \omega_d(1)$ be a size parameter. Let $\mathcal{C}$ be a class of compressible $L$-hidden-layer $\mathrm{poly}(d)$-sized ReLU networks mapping $\mathbb{Z}_q^d$ to $[0,1]$, and let $\mathcal{C}^\triangle$ be the lifted class of $(L+1)$-hidden-layer $d^{\Theta(m)}$-sized ReLU networks corresponding to $\mathcal{C}$, mapping $\mathbb{R}^d$ to $\mathbb{R}$ (as in Theorem 2.3). Suppose there is an SQ learner $A$ capable of learning $\mathcal{C}^\triangle$ over $\mathcal{N}(0, \mathrm{Id}_d)$ up to squared loss $d^{-\Theta(m)}$ using queries of tolerance $\tau$, where $\tau \geq d^{-\Theta(m^2)}$. Then there is an SQ learner $B$ that, using the same number of queries of tolerance $\tau/2$, produces a weak predictor $\widetilde{B}$ for $\mathcal{C}$ over $\mathrm{Unif}(\mathbb{Z}_q^d)$ with advantage $d^{-\Theta(m)}$ over guessing the constant $1/2$ (in expectation over both the data and the internal randomness of $\widetilde{B}$).*

*Proof of Theorem 3.1.* Let $m = m(d) = \log^c d$ for $c = \frac{1}{\alpha} - 1$, and let $d' = d^m = 2^{\log^{c+1} d}$, so that $d = 2^{\log^{1/(1+c)} d'}$. It is easy to see that the class $\mathcal{C}$ of parities on $\{0,1\}^d$ can be implemented by compressible one-hidden-layer $\mathrm{poly}(d)$-sized ReLU networks. Indeed, for any $S \subseteq [d]$ recall that $\chi_S(x) = \mathbb{1}(\sum_{j \in S} x_j \text{ is odd})$, which is a compressible one-hidden-layer network with the inner depth-0 network being $x \mapsto \sum_{j \in S} x_j$ and $\sigma(t) = \mathbb{1}[t \text{ is odd}]$. Thus the lifted class $\mathcal{C}^\triangle$ can be implemented by two-hidden-layer $d^{\Theta(m)}$-sized ReLU networks over $\mathbb{R}^d$. A padding argument lets us embed these classes into dimension $d'$. By using the predictor from Theorem 3.2 (with $q = 2$), we obtain an SQ algorithm capable of distinguishing parities from random labels using queries of tolerance $\tau/2$, assuming $\tau \geq d^{-\Theta(m^2)} = 2^{-\log^{2c+1} d} = 2^{-\log^{\frac{2c+1}{c+1}} d'}$. It is well-known [Kea98, BFJ+94] that the lower bound for learning parities is $\Omega(2^d \tau^2)$, which becomes $\Omega(2^{2^{\log^{1/(1+c)} d'}} \tau^2)$. Substituting $\alpha = \frac{1}{1+c}$ gives the result. $\qquad\square$

By way of an alternative construction that arguably remains hard even for non-SQ algorithms, in Appendix F we provide a different proof of this SQ lower bound using the LWR functions in place of the parities. We stress that this alternative proof remains unconditional and relies only on the LWR function family, *not* on the LWR hardness assumption itself.

# 4   Cryptographic Hardness Based on LWR

In this section we show hardness of learning two-hidden-layer ReLU networks over Gaussian inputs based on LWR. This is a direct application of the compressed DV lift (Theorem 2.3) to the LWR problem, which is by definition a hard learning problem over $\mathrm{Unif}(\mathbb{Z}_q^d)$.

**Theorem 4.1.** *Let $n$ be the security parameter, and fix moduli $p, q \geq 1$ such that $p, q = \mathrm{poly}(n)$ and $p/q = \mathrm{poly}(n)$. Let $d = n$. Let $c > 0$, $m = m(d) = \log^c d$ and $d' = d^m$. Suppose there exists a $\mathrm{poly}(d')$-time algorithm capable of learning $\mathrm{poly}(d')$-sized depth-2 ReLU networks under $\mathcal{N}(0, \mathrm{Id}_{d'})$ up to squared loss $1/\mathrm{poly}(d')$. Then there exists a $\mathrm{poly}(d') = 2^{\Theta(\log^{1+c} n)}$ time algorithm for $\mathsf{LWR}_{n,p,q}$.*

*Proof.* We claim that the class $\mathcal{C}_{\mathsf{LWR}}$ is implementable by compressible $\mathrm{poly}(d)$-sized one-hidden-layer ReLU networks over $\mathbb{Z}_q^d$, or, after padding, over $\mathbb{Z}_q^{d'}$. Indeed, by definition we have $f_w(x) = \frac{1}{p}\lfloor (w \cdot x) \bmod q \rfloor_p$, which is a compressible one-hidden-layer ReLU network with the inner depth-0 network (i.e., affine function) being $w \mapsto w \cdot x$ and $\sigma(t) = \frac{1}{p}\lfloor t \bmod q \rfloor_p$. Let $\mathcal{C}_{\mathsf{LWR}}^\triangle$ denote the corresponding lifted class of $\mathrm{poly}(d')$-sized two-hidden-layer ReLU networks, padded to have domain $\mathbb{R}^{d'}$. Applying Corollary C.6 to the assumed learner for $\mathcal{C}_{\mathsf{LWR}}^\triangle$, we obtain a $\mathrm{poly}(d')$-time weak predictor predictor for $\mathcal{C}_{\mathsf{LWR}}$, which readily yields a corresponding distinguisher for the $\mathsf{LWR}_{n,p,q}$ problem. Using the facts that $d' = d^m = 2^{\log^{1+c} d}$ and $d = n$, we may translate $\mathrm{poly}(d')$ into $2^{\Theta(\log^{1+c} n)}$, yielding the result. $\qquad\square$

*Remark* 4.2. Note that the choice of $m = m(d) = \log^c d$ in Theorem 4.1 is purely for simplicity. By picking $m(d) = \omega_d(1)$ to be a suitably slow-glowing function of $d$, such as $\log^* d$, we can obtain a running time for the final LWR algorithm that is as mildly superpolynomial as we like.

In addition, we also obtain a hardness result for one-hidden-layer networks under $\mathrm{Unif}\{0,1\}^d$, improving on the hardness result of [DV21] (see Theorem 3.4 therein) for *two*-hidden-layer networks under $\mathrm{Unif}\{0,1\}^d$. For this application, we let $d = n \log q = \widetilde{O}(n)$, so that we may identify the domain $\mathbb{Z}_q^n$ with $\{0,1\}^d$ via the binary representation. This also identifies $\mathrm{Unif}(\mathbb{Z}_q^n)$ with $\mathrm{Unif}\{0,1\}^d$.

**Corollary 4.3.** *Let $n, p, q$ be such that $p, q = \mathrm{poly}(n)$ and $p/q = \mathrm{poly}(n)$, and let $d = n \log q = \widetilde{O}(n)$. Suppose there exists an efficient algorithm for learning $\mathrm{poly}(d)$-sized one-hidden-layer ReLU networks under $U_d$ up to squared loss $1/4$. Then there exists an efficient algorithm for $\mathsf{LWR}_{n,p,q}$.*

## 5    Hardness of Learning using Label Queries

Here we show hardness of learning constant-depth ReLU networks over Gaussians from label queries by lifting pseudorandom function (PRF) families. For preliminaries on PRFs and their connection to hardness of learning, see Appendix A.4. Since PRFs are not necessarily compressibile, we will simply use the original DV lift (Theorem B.1).

**Theorem 5.1.** *Assume there exists a family of PRFs mapping $\{0,1\}^d$ to $\{0,1\}$ implemented by $\mathrm{poly}(d)$-sized $L$-hidden-layer ReLU networks. Then there does not exist an efficient learner that, given query access to an unknown $\mathrm{poly}(d)$-sized $(L+2)$-hidden-layer ReLU network $f : \mathbb{R}^d \to \mathbb{R}$, is able to output a hypothesis $h : \mathbb{R}^d \to \mathbb{R}$ such that $\mathbb{E}_{z \sim \mathcal{N}(0,\mathrm{Id}_d)}[(h(z) - f(z))^2] \leq 1/16$.*

*Proof.* Let $f_s : \{0,1\}^d \to \{0,1\}$ be an unknown $L$-hidden-layer ReLU network obtained from the PRF family by picking the key $s$ at random. Consider the lifted $(L+2)$-hidden-layer ReLU network $f_s^{\mathsf{DV}} : \mathbb{R}^d \to \mathbb{R}$ from Eq. (10), given by $f_s^{\mathsf{DV}}(z) = \mathrm{ReLU}(f_s(N_1(z)) - N_2'(z))$, where $N_1$ and $N_2$ are from Lemmas 2.4 and 2.5, and $N_2'(z) = \sum_j N_2(z_j)$. Suppose there were an efficient learner $A$ capable of learning functions of the form $f_s^{\mathsf{DV}}$ using queries. By the DV lift (Theorem B.1), $A$ yields an efficient predictor $B$ achieving small constant error w.r.t. the unknown $f_s$, contradicting Lemma A.3. We only need to verify that $A$'s query access to $f_s^{\mathsf{DV}}$ can be simulated by $B$. Indeed, suppose $A$ makes a query to $f_s^{\mathsf{DV}}$ at a point $z \in \mathbb{R}^d$. Then $B$ can make a query to $f_s$ at the point $\mathrm{sign}(z)$ and return $\mathrm{ReLU}(f_s(\mathrm{sign}(z)) - N_2'(z)) = f_s^{\mathsf{DV}}(z)$, as this was the key property satisfied by $f_s^{\mathsf{DV}}$. This completes the reduction and proves the theorem. $\square$

## Acknowledgments and Disclosure of Funding

We would like to thank our anonymous reviewers for pointing out an issue in the first version of our proof. Part of this work was completed while the authors were visiting the Simons Institute for the Theory of Computing. SC is supported in part by NSF Award 2103300. AG and ARK are supported by NSF awards AF-1909204, AF-1717896, and the NSF AI Institute for Foundations of Machine Learning (IFML). RM is supported by NSF CAREER Award CCF-1553605.

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
