# A    Technical preliminaries

## A.1    Notation

We use $\mathrm{Unif}(S)$ to denote the uniform distribution over a set $S$. We use $U_d$ as shorthand for $\mathrm{Unif}\{0,1\}^d$. We use $\mathcal{N}(0,\mathrm{Id}_d)$ (or sometimes $\mathcal{N}_d$ for short) to denote the standard Gaussian, and $|\mathcal{N}(0,\mathrm{Id}_d)|$ (or $|\mathcal{N}_d|$ for short) to denote the positive standard half-Gaussian (i.e., $g \sim |\mathcal{N}(0,\mathrm{Id}_d)|$ if $g = |z|$ for $z \sim \mathcal{N}(0,\mathrm{Id}_d)$). We use $[n]$ to denote $\{1,\ldots,n\}$.

For $q > 0$, $\mathbb{Z}_q$ will denote the integers modulo $q$, which we will identify with $\{0,\ldots,q-1\}$. We use $\mathbb{Z}_q/q$ to denote $\{0,1/q,\ldots,(q-1)/q\}$. Our discrete functions will in general have domain $\mathbb{Z}_q^d$ for some $q$. The $q = 2$ case, namely Boolean functions, have domain $\{0,1\}^d$. For the purposes of this paper, $\mathrm{sign} : \mathbb{R} \to \{0,1\}$ is defined as $\mathrm{sign}(t) = \mathbb{1}[t > 0]$. We will extend this to $\mathbb{Z}_q$ by defining $\mathrm{thres}_q : \mathbb{R} \to \mathbb{Z}_q$ in terms of a certain partition of $\mathbb{R}$ into $q$ intervals $I_0,\ldots,I_{q-1}$ (formally defined later) as the piecewise constant function that takes on value $k$ on $I_k$ for each $k \in \mathbb{Z}_q$. Scalar functions and scalar arithmetic applied to vectors act elementwise. We say a quantity is *negligible* in a parameter $n$, denoted $\mathrm{negl}(n)$, if it decays as $1/n^{\omega(1)}$.

A *one-hidden-layer* ReLU network mapping $\mathbb{R}^d$ to $\mathbb{R}$ is a linear combination of ReLUs, that is, a function of the form

$$F(x) = W_1\,\mathrm{ReLU}\left(W_0 x + b_0\right) + b_1,$$

where $W_0 \in \mathbb{R}^{k \times d}$, $W_1 \in \mathbb{R}^{1 \times k}$, $b_0 \in \mathbb{R}^k$, and $b_1 \in \mathbb{R}$. A *two-hidden-layer* ReLU network mapping $\mathbb{R}^d$ to $\mathbb{R}$ is a linear combination of ReLUs of one-hidden-layer networks, that is, a function of the form

$$F(x) = W_2\,\mathrm{ReLU}\left(W_1\,\mathrm{ReLU}\left(W_0 x + b_0\right) + b_1\right) + b_2,$$

where $W_0 \in \mathbb{R}^{k_0 \times d}$, $W_1 \in \mathbb{R}^{k_1 \times k_0}$, $W_2 \in \mathbb{R}^{1 \times k_1}$, $b_0 \in \mathbb{R}^{k_0}$, $b_1 \in \mathbb{R}^{k_1}$, and $b_2 \in \mathbb{R}$. Our usage of the term *hidden layer* thus corresponds to a *nonlinear* layer.

## A.2    Learning models

Let $\mathcal{C}$ be a function class mapping $\mathbb{R}^d$ to $\mathbb{R}$, and let $\mathcal{D}$ be a distribution on $\mathbb{R}^d$. We consider various learning models where the learner is given access in different ways to labeled data $(x, f(x))$ for an unknown $f \in \mathcal{C}$ and must output a (possibly randomized) predictor that achieves (say) squared loss $\varepsilon$ for any desired $\varepsilon > 0$. In the traditional PAC model, access to the data is in the form of iid labeled examples $(x, f(x))$ where $x \sim \mathcal{D}$, and the learner is considered efficient if it succeeds using $\mathrm{poly}(d, 1/\epsilon)$ time and sample complexity. In the Statistical Query (SQ) model [Kea98, Rey20], access to the data is through an SQ oracle. Given a bounded query $\phi : \mathbb{R}^d \times \mathbb{R} \to [-1, 1]$ and a tolerance $\tau > 0$, the oracle may respond with any value $v$ such that $|v - \mathbb{E}_{x \sim \mathcal{D}}[\phi(x, f(x))]| \leq \tau$. A correlational query is one that is linear in $y$, i.e. of the form $\phi(x, y) = \widetilde{\phi}(x)y$ for some $\widetilde{\phi}$, and a correlational SQ (CSQ) learner is one that is only allowed to make CSQs. An SQ learner is considered efficient if it succeeds using $\mathrm{poly}(d, 1/\epsilon)$ queries and tolerance $\tau \geq 1/\mathrm{poly}(d, 1/\epsilon)$. Finally, in the label query model, the learner is allowed to request the value of $f(x)$ for any desired $x$, and is considered efficient if it succeeds using $\mathrm{poly}(d, 1/\epsilon)$ time and queries.

## A.3    Learning with Rounding

Here we provide some further details about $\mathsf{LWR}$ (Definition 1.2) and known hardness results.

*Remark* A.1. Traditionally the $\mathsf{LWR}$ problem is stated with labels lying in $\mathbb{Z}_p$ instead of $\mathbb{Z}_p/p$, although both are equivalent since the moduli $p, q$ may be assumed to be known to the learner. The choice of $\mathbb{Z}_p/p$ is simply a convenient way to normalize labels to lie in $[0, 1]$. For consistency, we similarly normalize $\mathsf{LWE}$ labels to lie in $\mathbb{Z}_q/q$.

It is known that $\mathsf{LWE}_{n,q,B}$ is as hard as worst-case lattice problems when $q = \mathrm{poly}(n)$ and $B = q/\mathrm{poly}(n)$ (see e.g. [Reg10, Pei16] for surveys). Yet this is not known to directly imply the hardness of $\mathsf{LWR}_{n,p,q}$ in the regime in which $p, q$ are both $\mathrm{poly}(n)$, which is the one we will be interested in as $p, q$ will dictate the size of the hard networks that we construct in the proof of our cryptographic lower bound.

Unfortunately, in this polynomial modulus regime, it is only known how to reduce from LWE to LWR *when the number of samples is bounded relative to the modulus* [AKPW13, BGM$^+$16]. For instance, the best known reduction in this regime obtains the following hardness guarantee:

**Theorem A.2** ([BGM$^+$16])**.** *Let $n$ be the security parameter, let $p, q \geq 1$ be moduli, and let $m, B \geq 0$. Assuming $q \geq \Omega(mBp)$, any distinguisher capable of solving* LWR$_{n,p,q}$ *using $m$ samples implies an efficient algorithm for* LWE$_{n,q,B}$.

For our purposes, Theorem A.2 is not enough to let us base our Theorem 1.3 off of LWE, as we are interested in the regime where the learner has an *arbitrary* polynomial number of samples.

LWR with polynomial modulus and arbitrary polynomial samples is nevertheless conjectured to be as hard as worst-case lattice problems [BPR12] and has already formed the basis for a number of post-quantum cryptographic proposals [DKRV18, CKLS18, BGML$^+$18, JZ16]. We remark that one piece of evidence in favor of this conjecture is a reduction from a less standard variant of LWE in which the usual discrete Gaussian errors are replaced by errors uniformly sampled from the integers $\{-q/2p, \ldots, q/2p\}$ [BGM$^+$16].

Note also that for our purposes we require *quasipolynomial*-time hardness (or $T(n)$-hardness for $T(n)$ being any other fixed, mildly superpolynomial function of the security parameter) of LWR. While slightly stronger than standard polynomial-time hardness, this remains a reasonable assumption since algorithms for worst-case lattice problems are still believed to require at least subexponential time.

## A.4 Pseudorandom functions and hardness of learning

We recall the classical connection between pseudorandom functions and learning from label queries (also known as membership queries in the Boolean setting), due to Valiant [Val84] (see e.g. [BR17, Proposition 12] for a modern exposition).

**Lemma A.3.** *Let $\mathcal{C} = \{f_s\}$ be a family of PRFs from $\{0, 1\}^d$ to $\{0, 1\}$ indexed by the key $s$. Then there cannot exist an efficient learner $L$ that, given query access to an unknown $f_s \in \mathcal{C}$, satisfies*

$$\mathbb{P}_{x,s}[L(x) = f_s(x)] \geq \frac{1}{2} + \frac{1}{\text{poly}(d)},$$

*where the probability is taken over the random key $s$, the internal randomness of $A$, and a random test point $x \sim \text{Unif}\{0, 1\}^d$.*

There exist multiple candidate constructions of PRF families in the class TC$^0$ of constant-depth Boolean circuits built with AND, OR, NOT and threshold (or equivalently majority) gates. Because the majority gate can be simulated by a linear combination of ReLUs similar to $N_1$ from Lemma 2.4, any TC$^0_L$ (meaning depth-$L$) function $f : \{0, 1\}^d \rightarrow \{0, 1\}$ may be implemented as a $\text{poly}(d)$-sized $L$-hidden-layer ReLU network (see e.g. [VRPS21, Lemma A.3][2]). Thus we may leverage the following candidate PRF constructions in TC$^0$ for our hardness result:

- PRFs in TC$^0_4$ based on the decisional Diffie-Hellman (DDH) assumption [KL01] (improving on [NR97]), yielding hardness for depth-6 ReLU networks
- PRFs in TC$^0$ based on Learning with Errors [BPR12, BP14], yielding hardness for depth-$O(1)$ ReLU networks

Note that depth 4 is the shallowest depth for which we have candidate PRF constructions based on widely-believed assumptions, and the question of whether there exist PRFs in TC$^0_3$ is a longstanding open question in circuit complexity [Raz92, HMP$^+$93, RR97, KL01]. Under less widely-believed assumptions, [BIP$^+$18] have also proposed candidate PRFs in ACC$^0_3$.

## A.5 Partial assignments

Let $\alpha \in \{0, 1, \star\}^d$ be a *partial assignment*. We refer to $S(\alpha) : \{i \in [d] : \alpha_i = \star\} \subseteq [d]$ as the set of *free variables* and $[d] \backslash S(\alpha)$ as the set of *fixed variables*. Given two partial assignments $\alpha, \beta$, let the

---

[2]Note that what the authors term a depth-$(L + 1)$ network is in fact an $L$-hidden-layer network in our terminology.

*resolution* $\alpha \searrow \beta$ denote the partial assignment $\gamma$ obtained by substituting $\alpha$ into $\beta$. That is,

$$\gamma_i = \begin{cases} \star & i \in S(\alpha) \cap S(\beta) \\ \beta_i & i \in [d] \backslash S(\beta) \\ \alpha_i & i \in S(\beta) \backslash S(\alpha) \end{cases}$$

In this case we say that $\gamma$ is a *refinement* of $\beta$ that is the *result of applying* $\alpha$. We write $\gamma \in \mathrm{App}(\alpha)$ to denote that $\gamma$ is a result of applying $\alpha$. Note that the set of refinements of $\beta$ consists of all $3^{|S(\beta)|}$ partial assignments $\gamma \in \{0, 1, \star\}^d$ which agree with $\beta$ on all fixed variables of $\beta$.

Given $\alpha$, let $w(\alpha)$ denote $|\{i : \alpha_i = 1\}|$, that is, the Hamming weight of its fixed variables. Note that $w(\alpha \searrow \beta) \le w(\alpha) + w(\beta)$.

Given a function $h : \mathbb{R}^d \to \mathbb{R}$ and partial assignment $\gamma$, we use $h_\gamma : \mathbb{R}^d \to \mathbb{R}$ to denote its partial restriction given by substituting in $\gamma_i$ into the $i$-th input coordinate if $\gamma_i \in \{0, 1\}$. Note that given two partial restrictions $\alpha, \beta$,

$$(h_\beta)_\alpha = h_{\alpha \searrow \beta} \tag{9}$$

We say that $\alpha$ is *sorted* if the restriction of $\alpha$ to its fixed variables is sorted in nonincreasing order, e.g. $\alpha = (1, \star, 1, \star, \star, 0, 0)$ is sorted, but $\alpha = (1, \star, 0, \star, \star, 0, 1)$ is not. Given $\alpha$ which is not necessarily sorted, denote its *sorting* by $\overline{\alpha}$. In general, we will use overline notation to denote sorted partial assignments.

# B  The original DV lift

In order to set the stage for our compressed DV lift, we briefly outline the idea of the original DV lift in the setting of Boolean functions ($q = 2$). The goal is to approximate any given $f \in \mathcal{C}$ by a ReLU network $f^{\mathsf{DV}} : \mathbb{R}^d \to \mathbb{R}$ in such a way that $f^{\mathsf{DV}}$ under $\mathcal{N}_d$ behaves similarly to $f$ under $U_d$. As a first attempt, one might consider the function $f^\star(z) = f(\mathrm{sign}(z))$ (also studied in [KK14]), where recall that $\mathrm{sign}(t) = \mathbb{1}[t > 0]$. We could implement the following reduction: given a random example $(x, y)$ where $x \sim U_d$ and $y = f(x)$, draw a fresh half-Gaussian $g \sim |\mathcal{N}_d|$ and output $((2x - 1)g, y)$ (where the arithmetic in defining the vector $(2x - 1)g$ is done elementwise). Since $2x - 1$ is distributed uniformly over $\{\pm 1\}^d$, the marginal is exactly $\mathcal{N}_d$, and the labels are consistent with $f^\star$ since $\mathrm{sign}((2x - 1)g) = x$ and so $f(\mathrm{sign}((2x - 1)g)) = f(x)$. However, the issue is that the $\mathrm{sign}$ function is discontinuous, and so $f^\star$ is not realizable as a ReLU network.

Daniely and Vardi address this concern by devising a clever construction for $f^{\mathsf{DV}}$ that interpolates between two desiderata:

- For all but a small fraction of inputs, an initial layer successfully "Booleanizes" the input. In this case, one would like $f^{\mathsf{DV}}(z)$ to simply behave as $f(\mathrm{sign}(z))$.
- For the remaining fraction of inputs, we would ideally like $f^{\mathsf{DV}}$ to output an uninformative value such as zero, but this would violate continuity of $f^{\mathsf{DV}}$.

The trick is to use a continuous approximation of the sign function, $N_1$, that interpolates linearly between 0 and 1 on an interval $[-\delta, \delta]$ (see Fig. 1a), and to *pair it* with a "soft indicator" function $N_2 : \mathbb{R} \to \mathbb{R}$ for the region where $N_1 \ne \mathrm{sign}$. Concretely, $N_2(t)$ is constructed as a one-hidden-layer ReLU network that (a) is always nonnegative, (b) equals 0 when $|t| \ge 2\delta$, and (c) equals 1 when $|t| \le \delta$ (see Fig. 1b). Now let $N_2'(z) = \sum_j N_2(z_j)$, and define

$$f^{\mathsf{DV}}(z) = \mathrm{ReLU}(f(N_1(z)) - N_2'(z)). \tag{10}$$

One can show that $f^{\mathsf{DV}}$ satisfies $f^{\mathsf{DV}}(z) = \mathrm{ReLU}(f(\mathrm{sign}(z)) - N_2'(z))$, since $N_2'$ "zeroes out" $f^{\mathsf{DV}}$ wherever $N_1 \ne \mathrm{sign}$ for any coordinate. This lets us perform the following reduction: given examples $(x, y)$ where $x \sim U_d$ and $y = f(x)$, draw a fresh $g \sim |\mathcal{N}_d|$ and output $(z, \widehat{y}) = ((2x - 1)g, \mathrm{ReLU}(y - N_2'((2x - 1)g)))$. The marginal is again $\mathcal{N}_d$, and the labels are easily seen to be consistent with $f^{\mathsf{DV}}$. Correctness of the reduction can be established by using Gaussian anticoncentration to argue that $f^{\mathsf{DV}}$ is a good approximation of $f$. Formally, one can prove the following theorem.

**Theorem B.1** (Original DV lift, implicit in [DV21]). *Let $\mathcal{C}$ be a class of $L$-hidden-layer $\mathrm{poly}(d)$-sized ReLU networks mapping $\{0, 1\}^d$ to $[0, 1]$. There exists a class $\mathcal{C}^{\mathsf{DV}}$ of $(L + 2)$-hidden-layer*

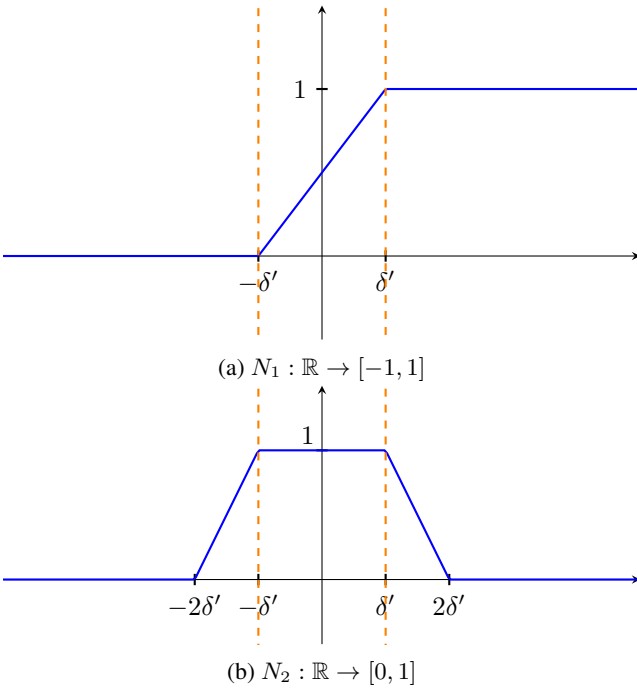

(a) $N_1 : \mathbb{R} \to [-1, 1]$

(b) $N_2 : \mathbb{R} \to [0, 1]$

Figure 1: Schematic plots of $N_1$ and $N_2$ in the $q = 2$ case, where $N_2'(z)$ may be realized as $\sum_{j \in [d]} N_2(z_j)$. Here, $\delta' = \Theta(\delta)$ where $\delta$ is the parameter from Lemmas 2.4 and 2.5.

poly$(d)$-*sized ReLU networks mapping* $\mathbb{R}^d$ *to* $[0, 1]$ *such that the following holds. Suppose there is an efficient algorithm $A$ capable of learning $\mathcal{C}^{DV}$ over $\mathcal{N}(0, \mathrm{Id}_d)$ up to squared loss $\frac{1}{64}$. Then there is an efficient algorithm $B$ capable of weakly predicting $\mathcal{C}$ over* $\mathrm{Unif}\{0, 1\}^d$ *with squared loss* $\frac{1}{16}$.

## C   Full proofs for Section 2

### C.1   Gadget constructions

We begin with the constructions of $N_1$ and $N_2$.

*Proof of Lemma 2.4.* This can be done by considering the piecewise linear function that approximates the function $\mathrm{thres}_q$ by matching it exactly on $\mathbb{R} \setminus \cup_k R_k$, and interpolating linearly between values $k - 1$ and $k$ on the interval $R_k$ for each $k \in [q - 1]$. $\qquad\square$

*Proof of Lemma 2.5.* Consider the piecewise linear function that is $0$ on $\mathbb{R} \setminus \cup_k S_k$, is $1$ on $\cup_k R_k$, and interpolates linearly between $0$ and $1$ (or $1$ and $0$) on $S_k \setminus R_k$ for every $k \in [q - 1]$. Put differently, the graph of $N_2$ consists of a trapezoid on each $S_k$ that achieves its maximum value of $1$ on $R_k$. $\qquad\square$

We now detail the construction of the gadget $N_3$. First let us briefly look at one approach that almost satisfies Eq. (2), except at the cost of exponential size. Let $\psi(s_1, \ldots, s_d; t)$ be any function that vanishes whenever $t \in \mathbb{Z} \setminus \{0\}$ (for all $s_1, \ldots, s_d \in [0, 1]^d$). For simplicity, let us consider the $d = 3$ case. Consider the following expression that resembles the inclusion-exclusion formula:

$$\psi(s_1, s_2, s_3; t) - \psi(1, s_2, s_3; t) - \psi(s_1, 1, s_3; t) - \psi(s_1, s_2, 1; t) \tag{11}$$
$$+ \psi(s_1, 1, 1; t) + \psi(1, s_2, 1; t) + \psi(s_1, 1, 1; t) - \psi(1, 1, 1; t)$$

Notice that whenever any $s_j = 1$, this expression vanishes identically. Moreover, for any $t \in \mathbb{Z} \setminus \{0\}$ (and any $s_1, \ldots, s_d$), the expression vanishes again because each summand vanishes. Thus the first two properties are satisfied; the third property turns out to be more subtle, and we will ignore it for the moment. The natural generalization of this expression to general $d$ can be stated in the language of partial assignments.

**Lemma C.1.** *Let $\psi : \mathbb{R}^d \to \mathbb{R}$ be any function. Let $\mathcal{P}_i$ denote the set of partial assignments $\gamma \in \{1, \star\}^d$ with $i$ 1s. The expression*

$$\sum_{i=0}^{d} \sum_{\gamma \in \mathcal{P}_i} (-1)^i \psi_\gamma \tag{12}$$

*vanishes whenever any $s_j = 1$. (We may view $t$ as an additional parameter that is always left free, as in Eq. (11))*

*Proof.* For concreteness, suppose $s_1 = 1$. Let $\mathcal{P}_i^\star$ (resp. $\mathcal{P}_i^1$) denote the set of $\gamma \in \mathcal{P}_i$ with $s_1 = \star$ (resp. $s_1 = 1$). For every $i \in \{0, \dots, d-1\}$, we can form a bijection between $\mathcal{P}_i^\star$ and $\mathcal{P}_{i+1}^1$ using the map $\gamma \mapsto \gamma'$ where $\gamma' = (1, \gamma_2, \dots, \gamma_d)$. When $s_1 = 1$, for every such pair $(\gamma, \gamma')$, we have $\psi_\gamma = \psi_{\gamma'}$, and moreover they occur in (12) with opposite signs. Thus the entire expression vanishes. $\qquad\square$

Let us assume for now that $\psi$ is picked suitably and the rest of the reduction goes through with this construction (as one can verify when we come to the proof of Theorem 2.3, this would indeed be the case). This construction has size $2^d$, meaning that the resulting lifted functions would have size $S = \text{poly}(2^d)$. But by Theorem F.5, the SQ lower bound for the LWR functions over $\mathbb{Z}_q^n$ with $n = d$ and $q = \text{poly}(n)$ scales as $q^{\Omega(n)} = 2^{\Omega(d \log d)} = S^{\Omega(\log \log S)}$, which is still superpolynomial in $S$. Thus after padding the dimension to $d' = 2^d$, this construction would actually still yield a superpolynomial SQ lower bound for two-hidden-layer ReLU networks over $\mathbb{R}^{d'}$.

Instead of pursuing this route, however, we give a more efficient construction that has size only slightly superpolynomial in $d$ by changing our goal to be closer to Eq. (3) (although still only approximately), while retaining the spirit of constructing $N_3$ using a linear combination of partial restrictions. The key lemma in proving Lemma 2.6 is the following.

**Lemma C.2.** *Let $m = m(d) = \omega_d(1)$ be a size parameter. Let $\mathcal{A}$ denote the set of all partial assignments $\alpha \in \{0, 1, \star\}^d$ for which $|S(\alpha)| = m$ and $w(\alpha) = 1$. Let $\mathcal{B}$ denote the set of all sorted partial assignments given by refining some element of $\mathcal{A}$ and sorting. Given $i, j \geq 0$, let $\mathcal{B}_{i,j}$ denote the set of $\overline{\beta} \in \mathcal{B}$ for which $|S(\overline{\beta})| = i$ and $w(\overline{\beta}) = j$. For any symmetric function $\psi : \mathbb{R}^d \to \mathbb{R}$, define the function*

$$\psi^* \triangleq \psi - \sum_{i=0}^{m} \sum_{j=1}^{m+1-i} (-1)^{m-i} \cdot \lambda_{i+j} \sum_{\overline{\beta} \in \mathcal{B}_{i,j}} \psi_{\overline{\beta}}, \qquad \text{for } \lambda_k \triangleq \binom{d-k-1}{m-k+1}$$

*Then*

(a) $|\mathcal{B}| \leq \binom{d}{m}(d-m) \cdot 3^m$
(b) $\psi^*$ *is symmetric*
(c) $\psi^*_\alpha : \mathbb{R}^d \to \mathbb{R}$ *is the identically zero function for all $\alpha \in \mathcal{A}$.*

*Proof.* Note that $|\mathcal{A}| = \binom{d}{m}(d-m)$. Any partial assignment $\beta$ has at most $3^{|S(\beta)|}$ refinements, and $\mathcal{B}$ is a subset of all refinements of partial assignments from $\mathcal{A}$, so $|\mathcal{B}| \leq \binom{d}{m}(d-m) \cdot 3^m$.

For the remaining parts of the lemma, it will be useful to observe that $\mathcal{B}$ consists exactly of all partial assignments with $i$ free variables and $j$ 1s for any $0 \leq i \leq m$ and $j \geq 1$ satisfying $i + j \leq m + 1$.

To prove the second part of the lemma, it suffices to show that

$$\sum_{\overline{\beta} \in \mathcal{B}_{i,j}} h_{\overline{\beta}} \tag{13}$$

is symmetric for all $i, j$. As transpositions generate the symmetric group on $d$ elements, it suffices to show that (13) is invariant under swapping two input coordinates, call them $a, b \in [d]$. For all $\overline{\beta} \in \mathcal{B}_{i,j}$ for which $a, b$ are either both present or both absent in $S(\overline{\beta})$, this clearly does not affect the value of $h_{\overline{\beta}}$. Now consider the set $S_a$ (resp. $S_b$) of partial assignments $\overline{\beta} \in \mathcal{B}_{i,j}$ for which only $a$ (resp. only $b$) is present in $S(\overline{\beta})$. There is a clear bijection $f : S_a \to S_b$: given $\overline{\beta} \in S_a$, swap the

$a$- and $b$-th entries, and vice-versa, and for any $\overline{\beta} \in S_a$, the function $h_{\overline{\beta}} + h_{f(\overline{\beta})}$ is unaffected by the swapping of input coordinates $a, b$. This concludes the proof of the second part of the lemma.

Finally, to prove the third part of the lemma, it suffices to verify it for a single $\alpha \in \mathcal{A}$, as $h^*$ is symmetric. So consider $\overline{\alpha} = \{1, 0, \cdots, 0, \star, \cdots, \star\}$. We apply (9) to get

$$h^*_{\overline{\alpha}} = h_{\overline{\alpha}} - \sum_{i=0}^{m} \sum_{j=1}^{m+1-i} (-1)^{m-i} \cdot \lambda_{i+j} \sum_{\overline{\beta} \in \mathcal{B}_{i,j}} h_{\overline{\alpha} \searrow \overline{\beta}}$$

$$= h_{\overline{\alpha}} - \sum_{\substack{\overline{\gamma} \in \mathcal{B} \cap \mathrm{App}(\alpha) \text{ sorted}}} h_{\overline{\gamma}} \cdot \sum_{i=0}^{m} \sum_{j=1}^{m+1-j} (-1)^{m-i} \cdot \lambda_{i+j} \sum_{\overline{\beta} \in \mathcal{B}_{i,j}} \mathbb{1}[\overline{\overline{\alpha} \searrow \overline{\beta}} = \overline{\gamma}] \qquad (14)$$

Note that for $\overline{\gamma} = \overline{\alpha}$, the only $\overline{\beta} \in \mathcal{B}$ for which $\overline{\overline{\alpha} \searrow \overline{\beta}} = \overline{\gamma}$ is $\overline{\beta} = \overline{\alpha}$. Indeed, for $\overline{\beta}$ to be such that $\overline{\overline{\alpha} \searrow \overline{\beta}} = \overline{\alpha}$, it must have $S(\overline{\beta}) = S(\overline{\alpha})$ and exactly one 1, from which it follows that $\overline{\beta} = \overline{\alpha}$. Since $\overline{\alpha} \in \mathcal{B}_{m,1}$, its coefficient in (14) is given by

$$(-1)^{m-m} \cdot \lambda_{m+1} = 1,$$

and so the $h_\alpha$ in (14) cancels with the $\overline{\gamma} = \overline{\alpha}$-th summand in (14).

In the rest of the proof, we can thus focus on sorted $\overline{\gamma} \in \mathcal{B} \cap \mathrm{App}(\alpha) \backslash \{\overline{\alpha}\}$. Note that such $\overline{\gamma}$ satisfy

$$|S(\overline{\gamma})| < m. \qquad (15)$$

To see this, recall that any $\overline{\gamma} \in \mathcal{B}$ with $|S(\overline{\gamma})| = m$ must have exactly one 1, and since $\overline{\gamma} \in \mathrm{App}(\overline{\alpha})$ it must be that $\overline{\gamma}$ must have $S(\overline{\gamma}) = S(\overline{\alpha})$ and so $\overline{\gamma} = \overline{\alpha}$.

Observe that we must have $\overline{\gamma}_1 = 1$. Indeed, it cannot be 0 because $\overline{\gamma}$ is sorted and has at least one 1. It also cannot be $\star$. To see this, consider any $\overline{\beta}$ for which $\overline{\overline{\alpha} \searrow \overline{\beta}} = \overline{\gamma}$. If we had $\overline{\beta}_1 \neq \star$, then clearly $\overline{\gamma}_1 \neq \star$. If we had $\overline{\beta}_1 = \star$, then $(\overline{\alpha} \searrow \overline{\beta})_1 = 1$ (as $\overline{\alpha}_1 = 1$), so $\overline{\gamma} = \overline{\overline{\alpha} \searrow \overline{\beta}}$ must also have first entry given by 1.

We are now ready to calculate the coefficient of $h_{\overline{\gamma}}$ (for each $\overline{\gamma} \in \mathcal{B} \cap \mathrm{App}(\alpha) \backslash \{\overline{\alpha}\}$) in (14) by adding the coefficients of all the $\overline{\beta} \in \mathcal{B}$ for which

$$\overline{\overline{\alpha} \searrow \overline{\beta}} = \overline{\gamma}. \qquad (16)$$

First let us consider the contribution of $\overline{\beta} \in \mathcal{B}$ for which $\overline{\beta}_1 = 1$. Observe that such $\overline{\beta}$ must have exactly $w(\overline{\gamma})$ 1s. Furthermore, such a $\overline{\beta}$ is an element of $\mathcal{B}$ if and only if it has at most $m + 1 - w(\overline{\gamma})$ free variables, and the set of free variables in $\overline{\beta}$ must be $S(\overline{\gamma}) \cup V$ where $V$ is any subset of $[d] \backslash (\{1\} \cup S(\overline{\alpha}))$. The contribution of all such $\overline{\beta}$ to the coefficient of $h_{\overline{\gamma}}$ in (14) is thus

$$\sum_{i=|S(\overline{\gamma})|}^{m+1-w(\overline{\gamma})} (-1)^{m-i} \cdot \lambda_{i+w(\overline{\gamma})} \cdot \binom{d - m - 1}{i - |S(\overline{\gamma})|}, \qquad (17)$$

where here the index $i$ denotes the total number of free variables in $\overline{\beta}$, and the factor of $\binom{d-m-1}{i-|S(\overline{\gamma})|}$ is the number of ways to choose $V$.

It remains to consider the contribution from $\overline{\beta} \in \mathcal{B}$ for which $\overline{\beta}_1 \neq 1$. First note that clearly we cannot have $\overline{\beta}_1 = 0$, as $\overline{\beta}$ is sorted and has at least one 1 because it lies in $\mathcal{B}$. The only possibility is $\overline{\beta}_1 = \star$, which we split into two cases based on $w(\overline{\gamma})$.

**Case 1:** $w(\overline{\gamma}) = 1$. In this case, we claim that there are no $\overline{\beta} \in \mathcal{B}$ simultaneously satisfying (16) and $\overline{\beta}_1 = \star$. Suppose to the contrary. Then such a $\overline{\beta}_1$ must have at least one 1 in some other entry (as $\overline{\beta} \in \mathcal{B}$), but this would imply that the resolution $\overline{\alpha} \searrow \overline{\beta}$ has at least two 1s, a contradiction. The total coefficient of $h_{\overline{\gamma}}$ in this case is thus exactly given by (17). Upon substituting $w(\overline{\gamma}) = 1$, this simplifies to

$$\sum_{i=|S(\overline{\gamma})|}^{m+1-w(\overline{\gamma})} (-1)^{m-i} \cdot \lambda_{i+1} \cdot \binom{d - m - 1}{i - |S(\overline{\gamma})|} = \sum_{i=|S(\overline{\gamma})|}^{m+1-w(\overline{\gamma})} (-1)^{m-i} \cdot \binom{d - i - 2}{d - m - 2} \cdot \binom{d - m - 1}{i - |S(\overline{\gamma})|} = 0,$$

where in the last step we use Lemma C.3 (which we can apply because of (15)).

**Case 2: $w(\overline{\gamma}) > 1$.** Observe that we must have $w(\overline{\beta}) = w(\overline{\gamma}) - 1$ (as the only entry of $\overline{\alpha}$ equal to 1 is the first entry, and the first entry of $\overline{\beta}$ is $\star$). As $w(\overline{\gamma}) - 1 > 0$ in the current case, such a $\overline{\beta}$ is an element of $\mathcal{B}$ if and only if it has at most $m + 2 - w(\overline{\gamma})$ free variables, and the set of free variables in $\overline{\beta}$ must be $\{1\} \cup S(\overline{\gamma}) \cup V$ where $V$ is any subset of $[d] \setminus (\{1\} \cup S(\overline{\alpha}))$. Thus in this second case, the contribution of all $\overline{\beta}$ with $\overline{\beta}_1 = \star$ to the coefficient of $h_{\overline{\gamma}}$ in (14) is

$$\sum_{i=|S(\overline{\gamma})|+1}^{m+2-w(\overline{\gamma})} (-1)^{m-i} \cdot \lambda_{i+w(\overline{\gamma})-1} \cdot \binom{d-m-1}{i-|S(\overline{\gamma})|-1} = \sum_{j=|S(\overline{\gamma})|}^{m+1-w(\overline{\gamma})} (-1)^{m-j-1} \cdot \lambda_{j+w(\overline{\gamma})} \cdot \binom{d-m-1}{j-|S(\overline{\gamma})|},$$

(18)

where here the index $i$ denotes the total number of free variables in $\overline{\beta}$, the factor of $\binom{d-m-1}{i-|S(\overline{\gamma})|-1}$ is the number of ways to choose $V$ (note that $|V| = i - |S(\overline{\gamma})| - 1$), and in the second expression we made the change of variable $j = i - 1$. We conclude that in this case, the coefficient of $h_{\overline{\gamma}}$ in (14) is given by the sum of (17) and (18), which is 0.

Overall, we conclude that the entire RHS of (14) vanishes for $\alpha \in \mathcal{A}$, proving the third part of the lemma. $\qquad\square$

We are ready to formally construct $N_3$ and verify that it has the required properties, is of acceptable size, and that it takes on nonzero values on a significant part of its domain.

*Proof of Lemma 2.6.* Let

$$\psi(s_1, \ldots, s_d; t) = \sum_{i=1}^{d} \mathrm{ReLU}\left(t - \left(s_i - \frac{1}{d-1}\sum_{j \neq i} s_j\right)\right) - \mathrm{ReLU}(dt),$$

viewed as a function of $s_1, \ldots, s_d$ parameterized by $t$, and let $\psi^*$ be as defined in Lemma C.2. Define $N_3(s_1, \ldots, s_d; t) = \psi^*(s_1, \ldots, s_d; t)$.

Part (a) follows directly from Lemma C.2(c). Part (b) follows by verifying that for any $t \in \mathbb{Z} \setminus \{0\}$, $\psi(s_1, \ldots, s_d; t) = 0$ for any $s_1, \ldots, s_d \in [0,1]^d$; this means that $\psi^*$, which is a combination of partial restrictions of $\psi$, also vanishes for such $t$. First suppose that $t$ is a positive integer. Observe that $t \geq 1$ while $s_i - \frac{1}{d-1}\sum_{j \neq i} s_j \in [-1,1]$, so each ReLU in the definition of $\psi$ is activated and we get

$$\psi(s_1, \ldots, s_d; t) = \sum_{i=1}^{d}\left[t - \left(s_i - \frac{1}{d-1}\sum_{j \neq i} s_j\right)\right] - dt = -\sum_{i=1}^{d}\left(s_i - \frac{1}{d-1}\sum_{j \neq i} s_j\right) = 0.$$

Next suppose that $t$ is a negative integer. Then $t \leq -1$ while $s_i - \frac{1}{d-1}\sum_{j \neq i} s_j \in [-1,1]$, so each ReLU in the definition of $h$ is inactive and we get $\psi(s_1, \ldots, s_d; t) = 0$.

For part (c), observe that by the size bound in Lemma C.2(a) and the fact that $\psi$ contains $O(d)$ ReLUs, the size of $N_3$ may be bounded by

$$S \leq O(d) \cdot \left(\binom{d}{m}(d-m) \cdot 3^m + 1\right) \leq O(d)\left(\frac{d^{m+1} \cdot 3^m}{m!} + 1\right) \leq d^{m+2} \leq d^{2m}$$

for $m$ larger than some absolute constant.

It remains to prove part (d). For brevity, we will omit the parameter $t$ and just refer to $\psi(0, \ldots, 0, s; t)$ and $\psi^*(0, \ldots, 0, s; t)$ as $\psi(0, \ldots, 0, s)$ and $\psi^*(0, \ldots, 0, s)$. We first compute $\psi(0, \ldots, 0, s)$: for $s \in [0,1]$,

$$\psi(0, \ldots, 0, s) = \mathrm{ReLU}(-s) + (d-1)\mathrm{ReLU}(\frac{1}{d-1} \cdot s) = s.$$

Next, for any $\overline{\beta} \in \mathcal{B}$, if $w(\overline{\beta}) = j$ for some $0 \leq j \leq m+1$, then if $\overline{\beta}_d = \star$,

$$
\begin{aligned}
\psi_{\overline{\beta}}&(0, \ldots, 0, s)\\
&= \psi(\underbrace{1, \ldots, 1}_{j}, \underbrace{0, \cdots 0}_{d-j-1}, s)\\
&= j \cdot \mathrm{ReLU}\left(-1 + \frac{1}{d-1}(j-1+s)\right) + (d-j-1) \cdot \mathrm{ReLU}\left(\frac{1}{d-1} \cdot j + \frac{1}{d-1} \cdot s\right)\\
&\quad + \mathrm{ReLU}\left(-s + \frac{1}{d-1} \cdot j\right)\\
&= \frac{d-j-1}{d-1} \cdot (j+s) + \mathrm{ReLU}\left(-s + \frac{1}{d-1} \cdot j\right)
\end{aligned}
$$

Note that when $s \in [0, 1/(d-1)]$, because $j \geq 1$ (as $\overline{\beta} \in \mathcal{B}$) this simplifies to

$$
= \frac{(d-j-s)j}{d-1}.
$$

On the other hand, if $\overline{\beta}_d \in \{0, 1\}$, then

$$
\begin{aligned}
\psi_{\overline{\beta}}(0, \ldots, 0, s) &= \psi(\underbrace{1, \ldots, 1}_{j}, \underbrace{0, \cdots 0}_{d-j})\\
&= j \cdot \mathrm{ReLU}\left(-1 + \frac{1}{d-1}(j-1)\right) + (d-j) \cdot \mathrm{ReLU}\left(\frac{1}{d-1} \cdot j\right) = \frac{(d-j)j}{d-1}.
\end{aligned}
$$

As there are $\binom{d-1}{i-1}$ (resp. $\binom{d-1}{i}$) partial assignments in $\mathcal{B}_{i,j}$ for which $\overline{\beta}_d = \star$ (resp. $\overline{\beta}_d \in \{0, 1\}$), we can thus explicitly compute $h^*(0, \ldots, 0, s)$ for $s \in [0, 1/(d-1)]$ to be

$$
\psi(0, \ldots, 0, s) - \sum_{i=0}^{m} \sum_{j=1}^{m+1-i} (-1)^{m-i} \binom{d-i-j-1}{m-i-j+1} \left(\binom{d-1}{i-1} \cdot \frac{(d-j-s)j}{d-1} + \binom{d-1}{i} \cdot \frac{(d-j)j}{d-1}\right).
$$

By Lemma C.4, the double sum is equal to sero, so $h^*(0, \ldots, 0, s) = h(0, \ldots, 0, s) = s$ for $s \in [0, 1/(d-1)]$ as claimed. $\qquad\square$

## C.2 Supporting technical lemmas for Lemma 2.6

**Lemma C.3.** *For any $0 \leq S < m \leq d$,*

$$
\sum_{i=S}^{m} (-1)^{m-i} \binom{d-i-2}{d-m-2}\binom{d-m-1}{i-S} = 0. \tag{19}
$$

*Proof.* We will show that for any integers $j \geq \ell \geq 1$,

$$
\sum_{k=0}^{\ell} (-1)^k \binom{j-k}{\ell-1}\binom{\ell}{k} = 0. \tag{20}
$$

We would like to substitute $\ell = d - m - 1$ and $j = d - 2 - S$. Note that this is valid as we can assume without loss of generality that $d - m - 1 \geq 1$ (otherwise $\binom{d-m-1}{i-S} = 0$ on the right-hand side of (19)), and $j \geq \ell$ by our assumption that $S < m$. We conclude the identity

$$
0 = \sum_{k=0}^{d-m-1} (-1)^k \binom{d-2-S-k}{d-m-2}\binom{d-m-1}{k} = \sum_{i=S}^{d-m-1+S} (-1)^{i-S} \binom{d-i-2}{d-m-2}\binom{d-m-1}{i-S}, \tag{21}
$$

where the second step is by the change of variable $i = k + S$. If $d - m - 1 + S \geq m$, then note that all summands $m < i \leq d - m - 1 + S$ vanish because in that case $d - i - 2 < d - m - 2$ and so $\binom{d-i-2}{d-m-2} = 0$. If $d - m - 1 + S < m$, then note that all summands $d - m - 1 + S < i \leq m$ vanish

because in that case $d - m - 1 < i - S$ and so $\binom{d-m-1}{i-S} = 0$. We conclude that (21) is equal, up to a sign, to the left-hand side of (19), so we'd be done.

It remains to establish (20), which we do by following an argument due to [Ear19]. Observe that the left-hand side of (20) is simply counting via inclusion-exclusion the number of subsets of $\{1, \ldots, j\}$ of size $\ell - 1$ which contain $\{1, \ldots, \ell\}$. Indeed, the $k = 0$ summand counts all subsets of size $\ell - 1$. The $k = 1$ summands subtract out the contribution, for every $1 \leq x \leq \ell$, from the subsets of size $\ell - 1$ which contain $x$. The $k = 2$ summands add back the contribution, for every distinct $1 \leq x < y \leq \ell$, from the subsets of size $\ell - 1$ which contain both of $x, y$, etc. $\qquad\square$

**Lemma C.4.** *For any integers $m \geq 3$ and $a \in \{0, 1, 2\}$,*

$$\sum_{i=1}^{m} \sum_{j=1}^{m+1-i} (-1)^{m-i} \binom{d-i-j-1}{m-i-j+1} \binom{d-1}{i-1} \cdot j^a = \mathbb{1}[a=0]$$

$$\sum_{i=0}^{m} \sum_{j=1}^{m+1-i} (-1)^{m-i} \binom{d-i-j-1}{m-i-j+1} \binom{d-1}{i} \cdot j^a = 0$$

*Proof.* By taking $\ell = i + j$, we can rewrite these sums as

$$S_{a,m} \triangleq \sum_{\ell=2}^{m+1} \sum_{i=1}^{\ell-1} (-1)^{m-i} \binom{d-1-\ell}{m+1-\ell} \binom{d-1}{i-1} (\ell - i)^a$$

$$T_{a,m} \triangleq \sum_{\ell=1}^{m+1} \sum_{i=0}^{\ell-1} (-1)^{m-i} \binom{d-1-\ell}{m+1-\ell} \binom{d-1}{i} (\ell - i)^a$$

We proceed by induction on $m$. The base cases follow from a direct calculation. By the change of variable $\ell' = \ell - 1$, we can rewrite $S_{a,m+1}$ as

$$-\sum_{\ell'=1}^{m+1} \sum_{i=1}^{\ell'} (-1)^{m-i} \binom{d-1-\ell'}{m+1-\ell'} \binom{d-1}{i-1} (\ell' + 1 - i)^a$$

$$= -\sum_{\ell'=1}^{m+1} \sum_{i=1}^{\ell'} (-1)^{m-i} \binom{d-1-\ell'}{m+1-\ell'} \binom{d-1}{i-1} (\ell' - i)^a$$

$$- \sum_{\ell'=1}^{m+1} \sum_{i=1}^{\ell'} (-1)^{m-i} \binom{d-1-\ell'}{m+1-\ell'} \binom{d-1}{i-1} \sum_{b=0}^{a-1} \binom{a}{b} (\ell' - i)^b \qquad (22)$$

Note that the first term on the right-hand side differs from $S_{a,m}$ only in the summands given by $1 \leq i = \ell' \leq m + 1$, and those summands clearly vanish. We conclude that the first term on the right-hand side of (22) is exactly $S_{a,m}$. For the second term on the right-hand side of (22), the part coming from any $0 < b \leq a - 1$ is also zero, so we thus get

$$= S_{a,m} - \sum_{\ell'=1}^{m+1} \sum_{i=1}^{\ell'} (-1)^{m-i} \binom{d-1-\ell'}{m+1-\ell'} \binom{d-1}{i-1}$$

$$= S_{a,m} - S_{0,m} - \sum_{\ell'=1}^{m+1} (-1)^{m-\ell'} \binom{d-1-\ell'}{m+1-\ell'} \binom{d-1}{\ell'-1}$$

$$= S_{a,m} - 1 - \sum_{\ell'=1}^{m+1} (-1)^{m-\ell'} \binom{d-1-\ell'}{m+1-\ell'} \binom{d-1}{\ell'-1}$$

$$= S_{a,m} = \mathbb{1}[a=0], \qquad (23)$$

where the penultimate step follows e.g. by applying the identity in [PSP17]. This completes the induction for $S_{a,m}$.

For $T_{a,m}$, note that by the change of variable $i' = i + 1$,

$$
\begin{aligned}
T_{a,m} &= -\sum_{\ell=1}^{m+1}\sum_{i'=1}^{\ell}(-1)^{m-i'}\binom{d-1-\ell}{m+1-\ell}\binom{d-1}{i'-1}(\ell-i'+1)^a \\
&= -\sum_{\ell=2}^{m+1}\sum_{i'=1}^{\ell-1}(-1)^{m-i'}\binom{d-1-\ell}{m+1-\ell}\binom{d-1}{i'-1}(\ell-i'+1)^a - \sum_{\ell=1}^{m+1}(-1)^{m-\ell}\binom{d-1-\ell}{m+1-\ell}\binom{d-1}{\ell-1} \\
&= -\sum_{b=0}^{a}\binom{a}{b}S_{b,m} + 1 = 0,
\end{aligned}
$$

where in the second step we pulled out the summands corresponding to $i' = \ell$, in the third step we used (23), and in the last step we used that for $m \geq 3$, $S_{b,m} = \mathbb{1}[b \neq 0]$ for $0 \leq b \leq 2$. $\qquad\square$

### C.3   Full proof of Theorem 2.3

*Continued proof of Theorem 2.3.* Let us compute the probability mass of the "good region" $G$. For coordinates $j \in [d-1]$, note that $\mathbb{P}[N_2(z_j) = 0] = \mathbb{P}[z_j \notin \cup_k S_k] = 1 - 2\delta = 1 - d^{-\Theta(m)}$. For $z_d$, we need a lower bound on the probability that $N_2(z_d) \in (\frac{1}{2d}, \frac{1}{d})$. Consider the behavior of $N_2$ on just the interval $S_k$ that is closest to the origin (which will be $k = \lceil q/2 \rceil$): it changes linearly from 0 to 1 (and again from 1 to 0) on $S_k \setminus R_k$. It is not hard to see that $N_2$ takes values in $(\frac{1}{2d}, \frac{1}{d})$ on a $O(1/d)$ fraction of $S_k$. Since the Gaussian pdf will be at least some constant on all of $S_k$, the probability that $z_d$ lands in this fraction of $S_k$ is $\Omega(|S_k|/d) = \Omega(\delta/qd) \geq d^{-\Theta(m)}$. Overall, we get that

$$
\mathbb{P}[z \in R] = \mathbb{P}\left[N_2(z_d) \in \left(\frac{1}{2d}, \frac{1}{d}\right)\right]\prod_{j \in [d-1]}\mathbb{P}[N_2(z_j) = 0] \geq (1 - d^{-\Theta(m)})^{d-1}d^{-\Theta(m)} = d^{-\Theta(m)},
$$

which is still $1/\operatorname{poly}(S)$ and hence non-negligible in the size $S$ of the network.

The discrete learner $B$ can now adapt $\widehat{f}$ as follows. Given a fresh test point $x \sim \operatorname{Unif}(\mathbb{Z}_q^d)$, the learner forms $z$ such that for each coordinate $j \in [d]$, $z_j$ is drawn from $\mathcal{N}(0,1)$ conditioned on $z_j \in I_{x_k}$; for brevity, we shall denote the random variable $z$ conditioned on $x$ (formed in this way) by $z|x$. If $z \in G$, then $B$ predicts $\widehat{y} = \frac{\widehat{f}(z)}{N_2(z_d)}$ (recall that when $z \in z$, $N_2(z_d) > \frac{1}{2d}$), and otherwise it simply predicts $\widetilde{y} = \frac{1}{2}$. The square loss of this predictor is given by

$$
\begin{aligned}
\mathop{\mathbb{E}}_{x\sim\operatorname{Unif}(\mathbb{Z}_q^d)}[(\widehat{y} - f(x))^2] &= \mathop{\mathbb{E}}_{x}\mathop{\mathbb{E}}_{z|x}[(\widehat{y} - f(x))^2] \\
&= \mathop{\mathbb{E}}_{x,z|x}[(\widehat{y} - f(x))^2 \mid z \in G]\mathbb{P}[z \in G] + \mathop{\mathbb{E}}_{x,z|x}[(\widehat{y} - f(x))^2 \mid z \notin G]\mathbb{P}[z \notin G] \\
&= \mathop{\mathbb{E}}_{x,z|x}\left[\left(\frac{\widehat{f}(z)}{N_2(z_d)} - f(x)\right)^2 \mid z \in G\right]\mathbb{P}[z \in G] + \mathop{\mathbb{E}}_{x,z|x}\left[\left(\frac{1}{2} - f(x)\right)^2 \mid z \notin G\right]\mathbb{P}[z \notin G] \\
&= \mathop{\mathbb{E}}_{x,z|x}\left[\left(\frac{\widehat{f}(z)}{N_2(z_d)} - \frac{f^\Delta(z)}{N_2(z_d)}\right)^2 \mid z \in G\right]\mathbb{P}[z \in G] + \mathop{\mathbb{E}}_{x}\left[\left(\frac{1}{2} - f(x)\right)^2\right]\mathbb{P}[z \notin G] \\
&\qquad\qquad\text{(by Eq. (8), when } z \in G, f^\Delta(z) = f(x)N_2(z_d)) \\
&< 4d^2\,\mathop{\mathbb{E}}_{z}[(\widehat{f}(z) - f^\Delta(z))^2 \mid z \in G]\mathbb{P}[z \in G] + \mathop{\mathbb{E}}_{x}\left[\left(\frac{1}{2} - f(x)\right)^2\right]\mathbb{P}[z \notin G] \\
&\qquad\qquad\text{(when } z \in G, N_2(z_d) > \frac{1}{2d}) \\
&\leq 4d^2\,\mathop{\mathbb{E}}_{z}[(\widehat{f}(z) - f^\Delta(z))^2] + \mathop{\mathbb{E}}_{x}\left[\left(\frac{1}{2} - f(x)\right)^2\right]\mathbb{P}[z \notin G] \\
&\leq 4d^2\varepsilon + \mathop{\mathbb{E}}_{x}\left[\left(\frac{1}{2} - f(x)\right)^2\right]\mathbb{P}[z \notin G] \\
&= \mathop{\mathbb{E}}_{x}\left[\left(\frac{1}{2} - f(x)\right)^2\right] + 4d^2\varepsilon - \mathop{\mathbb{E}}_{x}\left[\left(\frac{1}{2} - f(x)\right)^2\right]\mathbb{P}[z \in G].
\end{aligned}
$$

In the case of the hard classes $\mathcal{C}$ that we consider, we may assume without loss of generality that $\mathbb{E}_{x\sim\operatorname{Unif}(\mathbb{Z}_q^d)}[(\frac{1}{2} - f(x))^2] \geq 1/\operatorname{poly}(d)$, since otherwise the problem of learning $\mathcal{C}$ is trivial (in fact,

in our applications we will have $\mathbb{E}_{x \sim \mathrm{Unif}(\mathbb{Z}_q^d)}[(\frac{1}{2} - f(x))^2] = \Theta(1))$. This means that by taking

$$\varepsilon = \mathbb{P}[z \in G] / \mathrm{poly}(d) = d^{-\Theta(m)} / \mathrm{poly}(d) = d^{-\Theta(m)}$$

sufficiently small (but still $1/\mathrm{poly}(S)$), we may ensure that the square loss of the discrete learner $B$ is at most $\mathbb{E}_{x \sim \mathrm{Unif}(\mathbb{Z}_q^d)}[(\frac{1}{2} - f(x))^2] - d^{-\Theta(m)}$, as desired. □

*Remark* C.5. The only property of the Gaussian $\mathcal{N}(0, \mathrm{Id}_d)$ used crucially in the proof above is that it is a product distribution $P = \otimes_{i \in [d]} P_i$ where each $P_i$ is suitably anti-concentrated. By some simple changes to the parameters of $N_1$, $N_2$ and $N_3$ (i.e., adjusting the widths and locations of the intervals $I_k, R_k, S_k$ depending on each $P_i$), the proof can be made to work more generally for such distributions $P$.

**Corollary C.6** (Compressed DV lift with padding). *Let $q$, $m$ and $d$ be as above, and let $d' = d^m$. View $\mathcal{C}$ and $\mathcal{C}^\triangle$ as function classes on $\mathbb{Z}_q^{d'}$ and $\mathbb{R}^{d'}$ respectively, defined using only the first $d$ coordinates, so that $\mathcal{C}^\triangle$ is now a $\mathrm{poly}(d')$-sized class over $\mathbb{R}^{d'}$. Then an algorithm capable of learning $\mathcal{C}^\triangle$ over $\mathcal{N}_{d'}$ up to squared loss $1/\mathrm{poly}(d')$ implies a weak predictor for $\mathcal{C}$ over $\mathrm{Unif}(\mathbb{Z}_q^{d'})$ with advantage $1/\mathrm{poly}(d')$.*

# D  Barriers for constructing $N_3$

We briefly discuss why one natural approach to constructing $N_3$ satisfying the ideal properties in Eq. (2) ultimately requires two hidden layers rather than one, unlike the construction we give in Appendix C.

The most straightforward way to ensure that a function of $s_1, \ldots, s_d, t$ vanishes whenever there exists $j$ for which $s_j = 1$ would be to threshold on $\sum s_j$, e.g. by taking $\mathrm{ReLU}(1 - \sum_j s_j)$. While this function is a one-hidden-layer ReLU network, it is unclear how to modify it to satisfy the remaining desiderata in (2) while preserving the fact that it has only one hidden layer. We note that [DV21] takes this approach of thresholding on $\sum_j s_j$ but uses two hidden layers.

Here we informally argue that such an approach inherently requires an extra hidden layer. That is, we argue that no function $N : \mathbb{R}^2 \to \mathbb{R}$ that takes as inputs $s \triangleq \sum_j s_j$ and $t$ and satisfies (2) can be implemented as a one-hidden-layer network. Concretely, $N(s, t)$ must vanish whenever $s \geq 1$ or $t \in \mathbb{Z} \backslash \{0\}$. Any function computed by a one-hidden-layer ReLU network of the form $(s, t) \mapsto \sum_i \mathrm{ReLU}(a_i s + b_i t - c_i)$, unless if it is affine linear, must in general be nowhere smooth (i.e. have a discontinuous gradient) along the entire line where a particular neuron of the network vanishes. In our example, these are the lines $\{(s, t) : a_i s + b_i t = c_i\}$. But this means that such a line cannot intersect the region $\{(s, t) : s \geq 1\}$, as otherwise it would be zero (hence smooth) on an infinite segment of the line. This can only happen if $b_i = 0$, i.e. none of the neurons of $N$ depend on $t$. Such a network clearly cannot satisfy (2).

# E  Proof of Theorem 3.2

*Proof of Theorem 3.2.* Recall that $B$ is given SQ access to a distribution of pairs $(x, y)$ where $x \sim \mathrm{Unif}(Z_q^d)$ and $y = f(x)$ for an unknown $f \in \mathcal{C}$. $A$ can request estimates $\mathbb{E}[\phi(x, y)] \pm \tau$ for arbitrary bounded queries $\phi : \mathbb{Z}_q^d \times [0, 1] \to [-1, 1]$ and any desired $\tau$. We know that given $(x, y)$, the distribution of $(z, \widetilde{y})$, where $z = z(x)$ is defined by drawing each $z_j$ from $\mathcal{N}(0, 1)$ conditioned on $z_j \in I_{x_j}$ and $\widetilde{y} = \widetilde{y}(y, z)$ is as in Eq. (7)), is consistent with some $f^\triangle \in \mathcal{C}^\triangle$ except on a region of probability mass at most $d^{-9m^2}$ (recall Eq. (6)). Suppose we could simulate SQ access to the distribution of $(z, f^\triangle(z))$ using only SQ access to that of $(x, f(x))$. Then by the argument in Theorem 2.3, simulating $A$ on the $(z, f^\triangle(z))$ distribution would give us a weak predictor $\widetilde{B}$ for the distribution of $(x, f(x))$, satisfying

$$\mathbb{E}\left[\left(\widetilde{B}(x) - f(x)\right)^2\right] < \mathbb{E}\left[\left(\frac{1}{2} - f(x)\right)^2\right] - d^{-\Theta(m)}.$$

What we must describe is how $B$ can simulate $A$'s statistical queries. Say $A$ requests an estimate $\mathbb{E}_z[\phi(z, f^\triangle(z))] \pm \tau$ for some query $\phi : \mathbb{R}^d \times \mathbb{R} \to [-1, 1]$. Consider the query $\widetilde{\phi} : \mathbb{Z}_q^d \times [0, 1] \to$

$[-1,1]$ given by $\widetilde{\phi}(x,y) = \mathbb{E}_{z(x)}[\phi(z(x),\widetilde{y}(y,z(x)))]$. This function can be computed without any additional SQs, since the distribution of $(z,\widetilde{y}) = (z(x),\widetilde{y}(y,z(x)))$, given $(x,y)$, is fully determined and known to $B$. Observe that

$$\underset{x,y}{\mathbb{E}}\,\widetilde{\phi}(x,y) = \underset{x,z(x)}{\mathbb{E}}[\phi(z(x),\widetilde{y}(y,z(x)))] = \underset{z,\widetilde{y}}{\mathbb{E}}[\phi(z,\widetilde{y})]. \tag{24}$$

We must also account for the difference between $\mathbb{E}_z[\phi(z,f^{\triangle}(z))]$ and $\mathbb{E}_{z,\widetilde{y}}[\phi(z,\widetilde{y})]$. But because the distributions only differ on a region of mass $d^{-9m^2}$ and $\phi$ is bounded, we have

$$\left| \underset{z}{\mathbb{E}}[\phi(z,f^{\triangle}(z))] - \underset{z,\widetilde{y}}{\mathbb{E}}[\phi(z,\widetilde{y})] \right| \leq \Theta(d^{-9m^2}) \leq \frac{\tau}{2} \tag{25}$$

since we assumed $\tau \geq d^{-\Theta(m^2)}$. Putting together (24) and (25), we see that $B$ can simulate $A$'s query $\phi$ to within tolerance $\tau$ by querying $\widetilde{\phi}$ with tolerance $\tau/2$. $\qquad\square$

## F  SQ lower bound via the LWR functions

The SQ lower bound obtained via parities is somewhat unconvincing since there is a non-SQ algorithm capable of learning the lifted function class obtained from parities. Indeed, suppose we are given examples $(z, f^{\triangle}(z))$ where $f$ is an unknown parity. We know that whenever $z$ lands in the "good region" $G$ from the proof of Theorem 2.3 (which happens with non-negligible probability), we have $f^{\triangle}(z) = f(\text{sign}(z))N_2(z)$ (recall Eq. (8)). This means we can simply filter out all $z \notin G$ and form a clean data set of labeled points $(\text{sign}(z), f(\text{sign}(z)))$. The unknown $f$ (and hence $f^{\triangle}$) can now be learnt by simple Gaussian elimination.

In order to give a more convincing lower bound, we now provide an alternative proof of Theorem 3.1 using the LWR functions. The hard function class obtained this way is not only *unconditionally* hard for SQ algorithms, it is arguably hard for non-SQ algorithms as well, since LWR is believed to be cryptographically hard.

We begin by showing an SQ lower bound for the LWR functions using a general formulation in terms of pairwise independent function families. To our knowledge, this particular formulation has not appeared explicitly before in the literature, and was communicated to us by [Bog21]. A variant of this argument may be found in [BR17, §7.7].

**Definition F.1.** Let $\mathcal{C}$ be a function family mapping $\mathcal{X}$ to $\mathcal{Y}$, and let $D$ be a distribution on $\mathcal{X}$. We call $\mathcal{C}$ an $(1-\eta)$-pairwise independent function family if with probability $1-\eta$ over the choice of $x,x'$ drawn independently from $D$, the distribution of $(f(x),f(x'))$ for $f$ drawn uniformly at random from $\mathcal{C}$ is the product distribution $\text{Unif}(\mathcal{Y}) \otimes \text{Unif}(\mathcal{Y})$.

**Lemma F.2.** *Fix security parameter $n$ and moduli $p,q$. The $\mathsf{LWR}_{n,p,q}$ function class $\mathcal{C}_{\mathsf{LWR}} = \{f_w \mid w \in \mathbb{Z}_q^n\}$ is $(1 - \frac{2}{q^{n-1}})$-pairwise independent with respect to $\text{Unif}(\mathbb{Z}_q^n)$.*

*Proof.* This follows from the simple observation that whenever $x,x' \in \mathbb{Z}_q^n$ are linearly independent, the pair $(w \cdot x \bmod q, w \cdot x' \bmod q)$ for $w \sim \text{Unif}\{Z_q^n\}$ is distributed as $\text{Unif}(\mathbb{Z}_q) \otimes \text{Unif}(\mathbb{Z}_q)$. For such $x,x'$, $(f_w(x), f_w(x')) = (\frac{1}{p}\lfloor w \cdot x \bmod q \rfloor_p), \frac{1}{p}\lfloor w \cdot x' \bmod q \rfloor_p)$ for $f_w \sim \text{Unif}(\mathcal{C}_{\mathsf{LWR}})$ is distributed as $\text{Unif}(\mathbb{Z}_p/p) \otimes \text{Unif}(\mathbb{Z}_p/p)$. The probability that $x,x' \sim \text{Unif}(\mathbb{Z}_q^n)$ are linearly dependent is at most

$$\mathbb{P}[x=0] + \mathbb{P}[x \neq 0]\,\mathbb{P}[x' \text{ is a multiple of } x] \leq \frac{1}{q^n} + \frac{q}{q^n} \leq \frac{2}{q^{n-1}}.$$

$\qquad\square$

We can now prove full SQ lower bounds for any $(1-\eta)$-pairwise independent function family as follows.

**Lemma F.3.** *Let $\mathcal{C}$ mapping $\mathcal{X}$ to $\mathcal{Y}$ be a $(1-\eta)$-pairwise independent function family w.r.t. a distribution $D$ on $\mathcal{X}$. Let $\phi : \mathcal{X} \times \mathcal{Y} \to [-1,1]$ be any bounded query function. Then*

$$\underset{f \sim \text{Unif}(\mathcal{C})}{\text{Var}}\underset{x \sim D}{\mathbb{E}}[\phi(x,f(x))] \leq 2\eta.$$

*Proof.* Denote $\mathbb{E}_{x\sim D}[\phi(x, f(x))]$ by $\phi[f]$. By some algebraic manipulations (with all subscripts denoting independent draws),

$$
\begin{aligned}
\operatorname*{Var}_{f\sim\mathrm{Unif}(\mathcal{C})}[\phi[f]] &= \mathbb{E}_f\left[\phi[f]^2\right] - \left(\mathbb{E}_f[\phi[f]]\right)^2 \\
&= \mathbb{E}_f[\phi[f]\phi[f]] - \mathbb{E}_f[\phi[f]]\mathbb{E}_{f'}[\phi[f']] \\
&= \mathbb{E}_{f,f'}\left[\mathbb{E}_x[\phi(x, f(x))]\mathbb{E}_{x'}[\phi(x', f(x'))] - \mathbb{E}_x[\phi(x, f(x))]\mathbb{E}_{x'}[\phi(x', f'(x'))]\right] \\
&= \mathbb{E}_{x,x'}\mathbb{E}_{f,f'}\left[\phi(x, f(x))\phi(x', f(x')) - \phi(x, f(x))\phi(x', f'(x'))\right].
\end{aligned}
$$

By $(1-\eta)$-pairwise independence of $\mathcal{C}$, the inner expectation vanishes with probability $1-\eta$ over the choice of $x, x' \sim D$, and is at most 2 otherwise. This gives the claim. $\square$

**Theorem F.4.** *Let $\mathcal{C}$ mapping $\mathcal{X}$ to $\mathcal{Y}$ be a $(1-\eta)$-pairwise independent function family w.r.t. a distribution $D$ on $\mathcal{X}$. For any $f \in \mathcal{C}$, let $D_f$ denote the distribution of $(x, f(x))$ where $x \sim D$. Let $D_{\mathrm{Unif}(\mathcal{C})}$ denote the distribution of $(x, y)$ where $x \sim D$ and $y = f(x)$ for $f \sim \mathrm{Unif}(\mathcal{C})$ (this can be thought of as essentially $D \otimes \mathrm{Unif}(\mathcal{Y})$). Any SQ learner able to distinguish the labeled distribution $D_{f^*}$ for an unknown $f^* \in \mathcal{C}$ from the randomly labeled distribution $D_{\mathrm{Unif}(\mathcal{C})}$ using bounded queries of tolerance $\tau$ requires at least $\frac{\tau^2}{2\eta}$ such queries.*

*Proof.* Let $\phi : \mathcal{X} \times \mathcal{Y} \to [-1, 1]$ be any query made by the learner. For any $f \in \mathcal{C}$, let $\phi[f]$ denote $\mathbb{E}_{x\sim D}[\phi(x, f(x))] = \mathbb{E}_{(x,y)\sim D_f}[\phi(x, y)]$. Consider the adversarial strategy where the SQ oracle responds to this query with $\overline{\phi} = \mathbb{E}_{f\sim\mathrm{Unif}(\mathcal{C})}\phi[f] = \mathbb{E}_{(x,y)\sim D_{\mathrm{Unif}(\mathcal{C})}}[\phi(x, y)]$. By Chebyshev's inequality and Lemma F.3,

$$
\mathbb{P}_{f\sim\mathcal{C}}\left[\left|\phi[f] - \overline{\phi}\right| > \tau\right] \leq \frac{\mathrm{Var}_{f\sim\mathrm{Unif}(\mathcal{C})}\left[\phi[f]\right]}{\tau^2} \leq \frac{2\eta}{\tau^2}.
$$

So each such query only allows the learner to rule out at most a $\frac{2\eta}{\tau^2}$ fraction of $\mathcal{C}$. Thus to distinguish $D_{f^*}$ from $D_{\mathrm{Unif}(\mathcal{C})}$, the learner requires at least $\frac{\tau^2}{2\eta}$ queries. $\square$

Theorem F.5 now follows easily as a corollary.

**Theorem F.5.** *Let $\mathcal{C}_{\mathsf{LWR}}$ denote the $\mathsf{LWR}_{n,p,q}$ function class. Any SQ learner capable of learning $\mathcal{C}_{\mathsf{LWR}}$ up to squared loss $1/16$ under $\mathrm{Unif}(\mathbb{Z}_q^n)$ using queries of tolerance $\tau$ requires at least $\Omega(q^{n-1}\tau^2)$ such queries.*

*Proof.* It is not hard to see that learning $\mathcal{C}_{\mathsf{LWR}}$ up to squared loss $1/16$ certainly suffices to solve the distinguishing problem in Theorem F.4. The claim now follows by Lemma F.2. $\square$

We are ready for an alternative proof of Theorem 3.1.

*Alternative proof of Theorem 3.1.* Let $n$ be the security parameter, and fix moduli $p, q \geq 1$ such that $p, q = \mathrm{poly}(n)$ and $p/q = \mathrm{poly}(n)$. Let $d = n$, so that the SQ lower bound from Theorem F.5 is $\Omega(q^{n-1}) = d^{\Omega(d)} = 2^{\tilde{\Omega}(d)}$. Let $m = m(d) = \log^c d$ for $c = \frac{1}{\alpha} - 1$, and let $d' = d^m = 2^{\log^{c+1} d}$, so that $d = 2^{\log^{1/(1+c)} d'}$. Recall from the proof of Theorem 4.1 that the $\mathsf{LWR}_{n,p,q}$ function class $\mathcal{C}_{\mathsf{LWR}}$ is implementable by one-hidden-layer ReLU networks over $\mathbb{Z}_q^d$ of size $\mathrm{poly}(n) = \mathrm{poly}(d)$. The result now follows by Theorem 3.2 and the same padding argument as in the proof based on parities. $\square$