# OpenReview forum: "Hardness of Noise-Free Learning for Two-Hidden-Layer Neural Networks"
_NeurIPS.cc/2022/Conference — NeurIPS 2022 Accept_

### Official Review · Reviewer_ZuTs · 2022-06-30

**Rating:** 7
**Confidence:** 3
**Soundness:** 3 good
**Presentation:** 3 good
**Contribution:** 3 good

**Summary:**

The paper establishes hardness of learning neural networks with Gaussian inputs under various assumptions:
- statistical query learning, two hidden layers;
- cryptographic hardness of learning with rounding, two hidden layers;
- label query learning, existence of a family of pseudo-random functions, any fixed number of hidden layers.

The main tool for obtaining these results is a modified Danieli-Vardy transform (2021) that maps a Boolean example (x, y) to a Gaussian example (z, y') while remaining in the realizability setup.

**Questions:**

No questions

**Limitations:**

See weaknesses above.

**Strengths And Weaknesses:**

**Strengths**
- A solid theoretical work that establishes new hardness results for the now ubiquitous neural networks.

**Weaknesses**
- The paper does not explicitly indicate the limitations of the results. For example, lines 67-68 say that "Theorem 1.1 rules out almost all known approaches for provably learning neural networks", but the most well-known approach to learn neural networks---SGD---is not ruled out by Thm 1.1.

---

> ### Author Response · Authors · 2022-08-01
> **Thank you for the feedback**
>
> Thanks for the positive feedback! There is indeed a distinction between SQ and SGD, though note that by recent results of Abbe, Sandon, and coauthors, SGD with a suitable architecture + exotic initialization is essentially “P-complete,” so the only way to rule out SGD would be to show a computational lower bound by reducing from a complexity-theoretic conjecture. Fortunately, this is precisely what our lower bound in Theorem 1.3, based on LWR, achieves. We thus view our results as strong evidence that no efficient algorithm, not even SGD, can learn two hidden layer networks. We will tweak our language so this is clearer.

---

> > ### Comment · Reviewer_ZuTs · 2022-08-03
> > **Thank you for the clarification**
> >
> > I am satisfied with the author's response.

---

### Official Review · Reviewer_uQ7v · 2022-07-08

**Rating:** 8
**Confidence:** 3
**Soundness:** 4 excellent
**Presentation:** 4 excellent
**Contribution:** 3 good

**Summary:**

The authors prove statistical query lower bounds for learning polynomial-sized neural networks with two hidden layers. The bound is superpolynomial in the input dimension $d$ (or the query tolerance is negligible in $d$). No cryptographic assumptions are needed for these bounds to hold. The authors also show that, under the learning with rounding with polynomial modulus cryptographic assumption, no polynomial-time algorithm can learn neural networks with two hidden layers from Gaussian examples. The result is extended to neural networks with one hidden layer over the uniform distribution on the boolean hypercube.

**Questions:**

### Suggestions

A conclusion should be included in the main body were the paper to be accepted. If there is room, a table comparing different lower bounds setting and contrasting the current paper with related work would be a great addition.

**Limitations:**

Limitations: yes

Impact: N/A

**Strengths And Weaknesses:**

The paper is well-written and the review of the related work is thorough. A great deal of effort has been put into making sure that the paper is clear and accessible for a wide audience, particularly in the technical overview. This paper is a bit outside my expertise, but the results and techniques used are interesting and could be of independent interest. I believe this paper is relevant to the learning theory community.

---

> ### Author Response · Authors · 2022-08-01
> **Thank you for the feedback**
>
> Thanks for the very positive review! We will definitely try to add a conclusion and a table comparing the existing lower bounds in the final version of our paper if space allows.

---

### Official Review · Reviewer_PSRf · 2022-07-12

**Rating:** 7
**Confidence:** 5
**Soundness:** 4 excellent
**Presentation:** 4 excellent
**Contribution:** 4 excellent

**Summary:**

This work provides lower bounds in the noise free setting for learning two hidden layer networks in the Gaussian space. Basically, the whole concept is to embed hard problems over the uniform in hypercube to the Gaussian space. This is not something new, this has been done before in [1] for proving lower bound in the agnostic learning halfspaces over the Gaussian distribution. In contrast, this work provides noise-free lower bounds. They provide a super-polynomial SQ lower bound, a cryptographic lower bound under the LWR assumption. Moreover, they also provide lower bounds for the query model which is more powerful than the PAC learning model.

The whole concept is the following: To embed hard problems from the hypercube to the Gaussian space, we can use a similar idea like in [1], i.e., using the sign function. The DV lift basically does that by adding 2 more hidden layers (with ReLU components). So, a hard problem with L-Layers can provide lower bounds for L+2 layers in Gaussian space.

The authors decrease the number of layers from +2 Layers to +1. To do that, first they show a way to do it using a very large network, basically, they start from an exponential construction Eq.(11) and then they decrease it to $d^m$ by make the network more sparse, using the distributional properties of the Gaussian. They introduce some error in the construction but they show that this is indeed very small. After that the hardness proofs follow from a reduction.



[1] Adam Klivans and Pravesh Kothari. Embedding hard learning problems into gaussian space.

**Questions:**

Some presentation suggestions:
I would suggest the authors to add after section 1.2 lines 197 some more intuition about the construction and the idea that they use to decrease the size of the network.
Some fullstops and commas in equations are missing, i.e., eq. 2.

**Limitations:**

everything is good.

**Strengths And Weaknesses:**

# Pros
1. This is good result. The authors provide lower bounds under several assumptions/models.
2. This work is very well-written. Checked almost all the proofs and the claims are sound.

# Cons
Not really a con just a comment. The lower bounds are for 2-hidden layer networks where there are results for 1-hidden layer networks for the CSQ model. [GGJ+20],[DKKZ20] Basically, the trade-off is stronger model and 2-hidden layer instead of 1. In general, I would expect stronger lower bounds for 2-hidden layer network.


Overall, I recommend for acceptance.

---

> ### Author Response · Authors · 2022-08-01
> **Thank you for the feedback**
>
> Thanks for the positive feedback and for closely engaging with our work! We will make sure to incorporate the presentation suggestions. The comment on the tradeoff between expressivity of CSQ/SQ vs. complexity of the concept class is indeed a valid one, though establishing SQ lower bounds is significantly more challenging: whereas there are non-CSQ algorithms that circumvent the CSQ lower bounds for 1-hidden-layer [Chen-Klivans-Meka ‘20], there are no known efficient algorithms that circumvent our SQ lower bounds.

---

### Meta-Review · Area_Chair_T32s · 2022-08-26

**Recommendation:** Accept
**Confidence:** Certain

**Metareview:**

This work provides lower bounds in the noise free setting for learning two hidden layer networks in the Gaussian space. Overall it is a fundamental result well within the scope of Neurips, continuing a solid line of work and I cannot see any reason for rejection.

The authors have engaged with the reviewers, and have committed to make minor revisions and clarifications which I am sure they will do.

**Award:**

No

---

### Decision · Program_Chairs · 2022-09-14

Accept